# Bamboo for the Future: From Traditional Use to Industry 5.0 Applications

**DOI:** 10.3390/plants14193019

**Published:** 2025-09-29

**Authors:** Zishan Ahmad, Ritu Kumari, Bilal Mir, Taiba Saeed, Fatima Firdaus, Venkatesan Vijayakanth, Krishnamurthi Keerthana, Muthusamy Ramakrishnan, Qiang Wei

**Affiliations:** 1State Key Laboratory of Tree Genetics and Breeding, Co-Innovation Center for Sustainable Forestry in Southern China, Bamboo Research Institute, Key Laboratory of National Forestry and Grassland Administration on Subtropical Forest Biodiversity Conservation, School of Life Sciences, Nanjing Forestry University, Nanjing 210037, China; ahmad.lycos@gmail.com (Z.A.); vijayakanth2385@gmail.com (V.V.); keerthudaisy@gmail.com (K.K.); 2Department of Botany, School of Bioengineering and Biosciences, Lovely Professional University, Phagwara 144411, India; kritu8104@gmail.com (R.K.); bilalmir19933@gmail.com (B.M.); 3Department of Bioscience, Integral University, Lucknow 226026, India; taiba@iul.ac.in; 4Chemistry Department, Lucknow University, Lucknow 226007, India; fatimafirdaus74@gmail.com

**Keywords:** bamboo, cash crop, sustainability, green economy, bioeconomy, Industry 5.0, renewable resources

## Abstract

Bamboo (subfamily Bambusoideae, Poaceae) ranks among the fastest-growing plants on Earth, achieving up to 1 m day^−1^, significantly faster than other fast growing woody plant such as *Eucalyptus* (up to 0.6 m day^−1^) and *Populus* (up to 0.5 m day^−1^). Native to Asia, South America and Africa, and cultivated on approximately 37 million ha worldwide, bamboo delivers multifaceted environmental, social, and economic benefits. Historically central to construction, handicrafts, paper and cuisine, bamboo has evolved into a high-value cash crop and green innovation platform. Its rapid renewability allows multiple harvests of young shoots in fast-growing species such as *Phyllostachys edulis* and *Dendrocalamus asper*. Its high tensile strength, flexibility, and ecological adaptability make it suitable for applications in bioenergy (bioethanol, biogas, biochar), advanced materials (engineered composites, textiles, activated carbon), and biotechnology (fermentable sugars, prebiotics, biochemicals). Bamboo shoots and leaves provide essential nutrients, antioxidants and bioactive compounds with documented health and pharmaceutical potential. With a global market value exceeding USD 41 billion, bamboo demand continues to grow in response to the call for sustainable materials. Ecologically, bamboo sequesters up to 259 t C ha^−1^, stabilizes soil, enhances agroforestry systems and enables phytoremediation of degraded lands. Nonetheless, challenges persist, including species- and age-dependent mechanical variability; vulnerability to decay and pests; flammability; lack of standardized harvesting and engineering codes; and environmental impacts of certain processing methods. This review traces bamboo’s trajectory from a traditional resource to a strategic bioresource aligned with Industry 5.0, underscores its role in low-emission, circular bioeconomies and identifies pathways for optimized cultivation, green processing technologies and integration into carbon-credit frameworks. By addressing these challenges through innovation and policy support, bamboo can underpin resilient, human-centric economies and drive sustainable development.

## 1. Introduction

Bamboo is a highly versatile and rapidly growing plant, among the fastest growing on Earth [1]. It has been integral to the daily lives of millions in tropical regions, offering significant environmental, social, and economic benefits. Taxonomically, bamboo belongs to the subfamily Bambusoideae within the grass family Poaceae. From the dawn of civilization to the present day, bamboo has played a vital role in human life, earning it the moniker “the plant of multifunctional uses” [2,3,4]. Its rapid growth and early maturity allow for multiple harvests per year, offering a clear advantage over many traditional forest species. Bamboo thrives in regions receiving 1200 to 4000 mm of annual rainfall. Optimal growth temperatures vary among species: *Bambusa vulgaris* prefers average temperatures between 20 °C and 38 °C, while *Phyllostachys edulis* favors cooler climates with temperatures ranging from 18 °C to 26 °C. It can grow across a wide range of latitudes (from 50° N to 47° S) and altitudes (up to 4300 m above sea level), and adapts well to various soil types, including sandy, loamy, lateritic, and alluvial soils [5]. Native bamboo species are primarily found in Asia, South America, and Africa, while their presence in Europe, North America, and Australia is mainly due to introductions from these regions [5].

Bamboo shoots, whether fresh, dried, or pickled, have long been valued as health-promoting ingredients in traditional Asian cuisine. Modern processing technologies such as fermentation, roasting, boiling, blanching, canning, and pickling have further expanded their consumption as awareness of their nutritional benefits has increased [6,7]. In addition to its edible shoots, the bamboo stem is widely used as a raw material in housing, utensils, agriculture, handicrafts, packaging, and particularly in the paper and pulp industry, with approximately 35 bamboo species employed for this purpose [8]. The global bamboo market was valued at USD 67.13 billion in 2024 and is projected to grow at a compound annual growth rate of 4.7% between 2025 and 2030 (https://www.grandviewresearch.com; accessed on 22 August 2025). These figures reflect bamboo’s substantial economic importance across various industries, including construction, furniture, and textiles. Market expansion is expected to continue, driven by increasing demand for sustainable and eco-friendly materials.

Renowned for its flexibility and superior physical and mechanical properties, bamboo is increasingly regarded as a versatile crop rather than merely a forest product [9]. Its diverse applications are illustrated in Figure 1, which highlights both traditional and modern uses. Due to its rapid growth, high biomass production, and capacity to regenerate naturally after harvest, bamboo can be considered a valuable cash crop [10,11,12,13]. It provides a sustainable and consistent source of income for farmers. Moreover, the rising global demand for green products has enhanced its market value, making bamboo cultivation an attractive and profitable enterprise. Bamboo belongs to the subfamily *Bambusoideae* within the grass family *Poaceae*. Recent phylogenomic analyses have refined their classification into three distinct tribes: Bambuseae (tropical woody bamboos), encompassing approximately 79 genera and 1058 species; Arundinarieae (temperate woody bamboos), comprising around 37 genera and 618 species and Olyreae (herbaceous bamboos), including about 23 genera and 145 species [12,14,15].

Bambuseae also exhibits remarkable species diversity and ecological adaptability. Only a small portion of this is commonly grown and exploited for commercial purposes, although the genus *Bambusa* contains between 140 and 150 species [18]. Culm characteristics, ease of propagation, flowering cycle, stress tolerance, and end-use suitability are some of the factors that influence this selective usage. *Phyllostachys edulis*, *Bambusa balcooa*, *Dendrocalamus asper*, *Guadua angustifolia*, and *Bambusa vulgaris* are notable cultivated species that are used in a variety of industries, including food, paper, bioenergy, and construction [19,20]. Due to structural characteristics, local need, or propagation constraints, a number of species continue to be underutilised. Functional variation in features like shoot flavour, culm strength, and drought tolerance is further influenced by intraspecific variety, such as ecotypes and landraces; nevertheless, in many bamboo-growing countries, these elements are still little understood. A comprehensive grasp of the biological, agronomic, and molecular characteristics of various bamboo kinds is crucial for their focused cultivation, breeding, conservation, and industrial application in light of this wide genetic and species-level variation.

This review aims to provide a comprehensive synthesis of recent biological, physiological, agronomic, and genetic advances in bamboo research with a focus on stress responses, biotic constraints, and crop improvement prospects. It also emphasizes the importance of sustainable cultivation practices and modern applications, positioning bamboo as a future-ready resource aligned with global climate goals and the principles of Industry 5.0 [2].

## 2. Bamboo as a Cash Crop

Bamboo provides multiple income streams through its diverse applications in construction, furniture, textiles, and bioenergy. The rapid growth and early maturity of commercial species allow for frequent harvesting cycles, which can improve farm-level profitability [21]. Bamboo’s extensive rhizome-root systems improve soil health and prevent soil erosion, which contribute to sustainable land management in addition to its economic advantages. The demand for bamboo-based products is rising due to the increased focus on renewable and environmentally friendly materials worldwide, which makes growing bamboo an alluring and lucrative business venture for both farmers and industries [22]. For example, bamboo residues used in hard-carbon production yield minimum selling prices between 14 and 18 CNY/kg in China, and U.S. plantations for fiber production show farm-gate break-even prices of USD 48–55/ton with profitability by year 5 [13,23]. These values compare favorably with alternative crops such as maize (USD 35–40/ton) (https://ycharts.com/indicators/us_maize_price; accessed on 22 August 2025) or plantation timber (USD 30–45/ton), indicating that bamboo can achieve relatively higher market prices under specific production systems (http://tfsweb.tamu.edu/timberpricetrends/). However, comprehensive economic assessments, including cost–benefit analyses and market volatility studies, remain limited and should be further explored to better quantify bamboo’s role as a cash crop in different socio-economic contexts. Table 1 represents the comparative mechanical and environmental properties of bamboo and conventional materials.

It also plays a vital role in climate change mitigation by releasing more oxygen and absorbing more carbon dioxide than an equivalent stand of trees, thus significantly contributing to carbon sequestration [40,41]. Despite its advantages, bamboo cultivation and utilization face several challenges. These include limited awareness of its economic potential, lack of standardized harvesting and processing methods (leading to inconsistent product quality), competition from other materials, and the necessity for sustainable management practices to prevent overexploitation and ensure long-term viability [13]. This section highlights both the traditional and modern applications of bamboo, with a focus on its role in sustainable energy production and environmental solutions.

### 2.1. Bamboo as a Sustainable Energy and Environmental Solution

As a fast-growing plant with high biomass productivity, bamboo holds substantial promise as a renewable energy resource and environmental solution. It can be harvested sustainably and used as a feedstock to produce bioenergy products such as biofuels, biogas, and biochar [42]. These energy pathways reduce dependence on fossil fuels and help mitigate greenhouse gas emissions, thus supporting global climate goals. Given this potential, bamboo is cultivated and protected in various Asian, African, and South American countries as a strategic renewable resource [43]. Protection generally involves policy-driven measures such as restricting overharvesting, regulating land-use change, establishing bamboo reserves, and promoting community-based management practices [44,45]. In some regions, governments and NGOs also implement reforestation and restoration programs, provide incentives for sustainable harvesting, and support certification schemes to ensure long-term resource security. Its ecological benefits are noteworthy; bamboo absorbs significant amounts of carbon dioxide and releases more oxygen than conventional wood biomass. This is due in part to its efficient carbon storage in cell walls and its lignocellulosic composition, which exceeds 70% in most species. Notably, Moso bamboo (*Phyllostachys edulis*) contains up to 78% lignocellulose, making it a valuable source of natural biomass [46,47] (Figure 2).

Bamboo’s high biomass yield per unit area and extensive rhizome–root system facilitates efficient nutrient uptake and rapid growth, thereby enhancing its carbon sequestration potential [48]. Its regenerative capacity allows for repeated harvesting without killing the plant. Bamboo’s high cellulose and lignin content make it particularly suitable for biofuel production through various methods, including biodegradation [49,50], steam explosion [51], and acid-base pretreatment [52]. These processes can convert bamboo lignocellulose into alcohol [50], biogas [53], glucose [49], and bio-oil [54].

Bamboo has a lot of potential as a feedstock for biorefineries, but it is not yet always more competitive than more conventional biomass sources like straw or sugarcane bagasse [55]. Its complex lignocellulosic structure and high lignin content necessitate enzymatic hydrolysis and energy-intensive pretreatment, raising processing costs. Its economic competitiveness is further diminished by large culms, limited harvesting mechanisation, and logistical challenges in storage and transportation [56]. Therefore, improvements in processing technologies, efficient supply chains, and the development of infrastructure unique to a given region are necessary for the viability of using bamboo for the production of biofuel on a large scale [13].

**Figure 2 plants-14-03019-f002:**
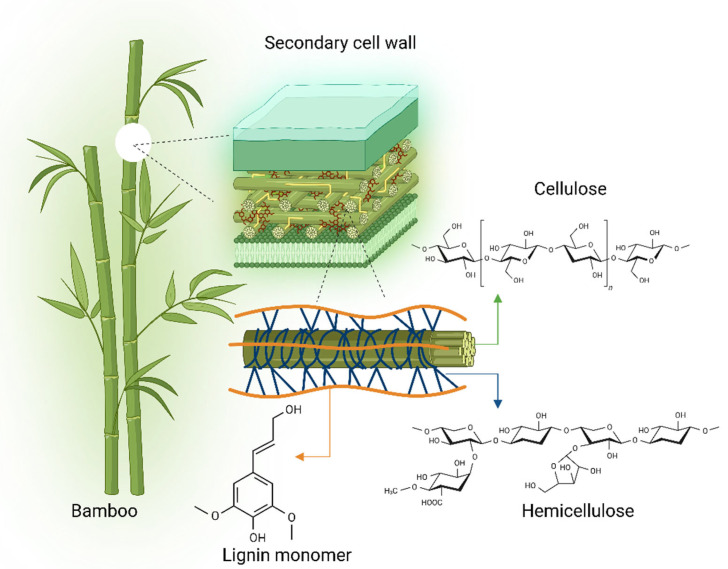
Structural composition of bamboo highlighting the secondary cell wall and its major lignocellulosic components. The secondary cell wall of bamboo is primarily composed of cellulose microfibrils embedded in a matrix of hemicellulose and lignin. Cellulose provides mechanical strength, while hemicellulose and lignin contribute to structural integrity and flexibility. The magnified schematic illustrates the organization of these components: cellulose chains, hemicellulose polymers, and lignin monomers [57]. Bamboo’s high lignocellulosic content makes it an efficient and renewable biomass source for applications in biofuel production, biocomposites, and sustainable material development [58].

Bamboo biomass can also be transformed into energy through combustion, gasification, or pyrolysis, depending on the desired output (heat, electricity, or liquid fuels) [41,59]. For example, anaerobic digestion of bamboo biomass by microorganisms produces biogas, a renewable energy source consisting mainly of methane and carbon dioxide [60]. This biogas can be used for cooking, heating, or electricity generation. Bamboo charcoal, produced via carbonization [20], is used for air and water purification due to its ability to absorb contaminants, pollutants, and odors. Bamboo can also be pelletized into a dense, low-emission solid biofuel with high energy content, which is easy to transport and store. Additionally, bamboo can be burned directly as a thermal energy source, suitable for both industrial and domestic applications. Cogeneration systems (combined heat and power, CHP) can also utilize bamboo biomass to generate both electricity and heat, increasing overall energy efficiency [60]. Bamboo residues, including leaves and branches, can be used in waste-to-energy systems, minimizing environmental waste and maximizing energy output.

Pyrolysis of bamboo produces biochar, a carbon-rich byproduct that enhances soil fertility, water retention, and long-term carbon sequestration. The pyrolysis process varies in efficiency depending on the composition of the bamboo’s cellulose, hemicellulose, and lignin components [61]. These components influence product yield and quality due to variations in chemical bonding and material structure during thermal conversion. Such flexibility makes bamboo a strong candidate for replacing traditional solid fuels and aligns it with current sustainable energy research priorities [62]. In addition to solid and gaseous fuels, bamboo can be converted into liquid biofuels such as bioethanol and biodiesel, offering promising alternatives for transportation fuels. Fermentation and enzymatic hydrolysis are used to break down bamboo biomass into fermentable sugars, which can then be converted into bioethanol. This ethanol can be blended with petrol to reduce emissions from transportation. Bamboo oil, extracted from seeds or processed biomass, can be transesterified to produce biodiesel, a renewable and cleaner-burning substitute for petroleum diesel.

### 2.2. Carbon Sequestration and Climate Change Mitigation

Bamboo forests have significant potential for carbon sequestration due to their rapid growth rates and strong annual regrowth after harvesting, especially when harvested culms are used to produce long-lasting products. Advances in processing technologies have improved the durability of bamboo-based goods, ensuring that the carbon stored within them remains sequestered for extended periods rather than quickly returning to the atmosphere (Figure 3).

In a typical Moso bamboo habitat, total carbon storage is estimated to range from 87.83 to 119.5 Mg C ha^−1^, encompassing both biomass and soil pools [63], with soil carbon comprising roughly two-thirds of the total. Annual carbon sequestration by Moso bamboo ranges from 6.0 to 7.6 Mg C ha^−1^ yr^−1^, underscoring its potential as a prime species for carbon fixation [63]. In a *P. bambusoides* stand in Japan, aboveground carbon stock was 52.3 Mg C ha^−1^, while belowground carbon, including 20.8 Mg C ha^−1^ in roots and 92.0 Mg C ha^−1^ in soil, totaled 112.8 Mg C ha^−1^ [64]. Similarly, in northwestern Ethiopia’s highland bamboo plantations, aboveground carbon stocks ranged from 87.5 Mg C ha^−1^ on riverbank sites to 111.6 Mg C ha^−1^ on homestead plots [65].

Due to its high annual carbon accumulation rate, bamboo is one of the most effective forest vegetation types for carbon fixation. Depending on calculation methods, the estimated total carbon stored in Chinese bamboo forests between 1999 and 2003 ranges from 605.5 to 1425 Tg C [66]. Since the 1950s, bamboo carbon stocks in China have increased dramatically, in parallel with the expansion of bamboo cultivation. For example, carbon storage rose from 318.6 Tg C (1950–1962) to 631.6 Tg C during 1999–2003. The carbon density of Chinese bamboo forests, ranging from 168.7 to 259.1 t C ha^−1^, is significantly higher than China’s average forest carbon density (38.7 t C ha^−1^) and even surpasses the global average of 86 t C ha^−1^ [67,68]. These values underscore bamboo’s extraordinary potential for carbon sequestration.

Looking ahead, these findings suggest that bamboo could be strategically developed and managed to maximize its role in climate change mitigation. Further research and innovation in bamboo cultivation and product development could enhance both durability and carbon retention. Additionally, expanding bamboo plantations, especially in areas well-suited to its growth, could provide substantial ecological and economic benefits. Integrating bamboo forestry into carbon offset markets and national climate strategies may be a promising path to harness its full potential in global climate action. These prospects position bamboo as a vital component in the transition toward sustainable and low-emission energy systems. Bamboo belongs to the Poaceae family, which also includes widely cultivated species such as maize, wheat, rye, oats, sugarcane, barley, and rice. There are approximately 1250 bamboo species across 75 genera worldwide [69]. According to FAO/INBAR, Asia contains the most bamboo resources, spanning over 10 million hectares, more than half of which (5.4 million ha) are found in China (INBAR 2021; https://www.inbar.int/; accessed on 22 August 2025) [70]. With a total export value of USD 2.47 billion, Asia-Pacific remained the world’s top exporter of bamboo products, according to INBAR (INBAR 2021). A total of USD 297 million was exported by Europe, followed by USD 265 million from North America and USD 19 million from Africa. With USD 882 million in imports, Europe was the largest importer. The import values for other regions, including South and Central America, Asia-Pacific, and North America, were USD 64 million, 812 million, and USD 836 million, respectively (INBAR 2021) [70]. In Brazil, the southwestern Amazon region is particularly rich in bamboo, containing around 18 million hectares of bamboo-dominated forest. Despite global declines in overall forest cover, the area under bamboo forests is steadily increasing at a rate of 3% per year [71]. This region harbors one of the world’s largest natural bamboo reserves, spanning approximately 180,000 km^2^ and supports 65% of all known bamboo species and 89% of bamboo genera present in North America, including *Dendrocalamus asper*, *Bambusa tuldoides*, and *B. vulgaris.*

Bamboo exhibits a greater ability to sequester carbon and be used in industry than other biomass resources. For instance, compared to major biofuel crops like maize (6.9–9.1 t C ha^−1^), sugarcane (11.2–13.5 t C ha^−1^), and sorghum (5.7–8.4 t C ha^−1^), the carbon density of Chinese bamboo forests (168.7–259.1 t C ha^−1^) is substantially higher [72]. Even woody crops that grow quickly, like poplar and willow, typically produce 30 to 50 t C ha^−1^, which is still far less than bamboo’s capacity to store carbon. In terms of productivity, bamboo’s annual biomass accumulation rate of 10–30 t ha^−1^ surpasses most cereal crops and rivals energy-dedicated plantations such as eucalyptus [73]. Bamboo’s perennial growth enables sustained carbon fixation with lower management costs and less soil degradation than maize or sugarcane, which need intensive inputs and annual replanting. From an industrial standpoint, bamboo has many uses that go beyond the production of energy and biofuel. These include textiles, bioplastics, durable building materials, and activated carbon, which prolongs the retention of carbon in final products. Going forward, bamboo’s distinct position between woody perennials and agricultural crops raises the possibility that it could replace or even supplement fossil fuel-intensive industries like cement and coal by offering a more carbon-efficient renewable feedstock. Over the coming decades, strategic investments in bamboo-based bioenergy and biomaterials, in addition to international carbon offset programs, could establish bamboo as a key component of sustainable resource economies. Although these figures demonstrate bamboo’s exceptional capacity to sequester carbon, it is crucial to remember that a thorough evaluation must also take processing, transportation, and end-of-life emissions into consideration. Therefore, it will be essential to use life cycle assessment (LCA) techniques to confirm the actual net climate benefits of bamboo in comparison to other biomass resources.

## 3. Nourishing and Healing: Bamboo’s Dual Role in Food and Pharmaceuticals

The bamboo plant, endowed with a plethora of functional components, is a remarkably valuable food source. Fresh bamboo shoots typically contain 1.49–4.04 g protein per 100 g fresh weight (≈1.5–4.0%) and crude fiber ranging from 7.3 to 12.8 g per 100 g (7.3–12.8%) [4,74,75]. According to mineral analyses, potassium can reach 632 mg per 100 g, but fat content is still low (less than 1%) [76]. It should be mentioned that these values vary significantly between species. *Bambusa stenoaurita*, for instance, has one of the highest fibre contents (~12.8%), whereas other species may have as low as 7.3% [74].

### 3.1. Edible Forms of Bamboo

Many animals consume young shoots, leaves, culms, and seeds of various bamboo species. For some, such as the giant panda and golden lemurs, bamboo is the primary food source, while for others like the mountain gorilla, it is a seasonal delicacy [20]. In parts of South India, the seeds of *B. arundinacea* and *P. bambusoides* are colloquially known as “bamboo rice” and are consumed similarly to traditional rice [77]. Historical records from the 1883 famine in India describe *B. arundinacea* producing a prolific seed yield, which served as an important food source across South Indian communities [77]. The consumption of bamboo shoots is most prevalent in Southeast and East Asia [78]. In India, bamboo is predominantly used in the Northeastern region, where it forms an essential component of traditional cuisine. Various ethnic communities consider fresh or fermented bamboo shoots a delicacy. Fresh shoots are crisp and crunchy, and are incorporated into soups, stir-fries, snacks, salads, fried rice, spring rolls, and other dishes [79]. Notable bamboo-based traditional foods include *ushoi*, *soibum*, *rep*, *mesu*, *eup*, *ekhung*, and *hirring* [80]. Shoots from species such as *P. pubescens*, *D. hamiltonii*, *D. giganteus*, *B. balcooa*, *B. bambos*, and *Thyrsostachys siamensis* are most commonly consumed, either fresh or fermented [81].

In Northeast India, fermented bamboo shoots are a dietary staple. A variety of fermentation techniques have evolved to match different bamboo species and regional tastes [82]. Fermentation has a significant role in prolonging the shelf life of bamboo shoots from approximately 2–3 days to over a year. In addition to preservation, this procedure drastically lowers the amounts of cyanogenic glycosides, such as taxiphyllin, which are found naturally in many bamboo species and can be harmful if improperly handled [83]. In one study, for instance, the cyanogenic (taxiphyllin) content decreased significantly throughout the course of 30 days of fermentation, from 434.9 ppm to 164.8 ppm, with the largest decrease taking place in the first 18 days. This brought the levels closer to the WHO-recommended ~10 ppm safety threshold [83]. Fermentation increases the nutritional quality and safety of bamboo shoots by reducing these anti-nutritional substances. It also raises the amounts of minerals, proteins, amino acids, and bioactive compounds, which adds to the health advantages of bamboo shoots [84]. Bamboo shoots are also valued for their rich dietary fiber content, which helps regulate cholesterol and supports overall nutritional health. They contain essential vitamins such as B6, B3, B1, A, and E [81]. The growth of a bamboo shoot involves the development of a juvenile culm composed of compact leathery sheaths enclosing internodes. These are typically harvested when they reach 15–16 cm in height. Once the fibrous sheath is removed, the inner culm is washed and processed for consumption. *P. edulis* and *P. pubescens* are commonly used in culinary preparations.

Young bamboo shoot culm flour from *Dendrocalamus* species has also shown promise in food applications. Analyses indicate that this flour contains 67–79 g of fiber per 100 g, along with elevated levels of protein, carbohydrates, and fats, surpassing standard nutritional benchmarks. These findings support its potential as a viable alternative food source. Because bamboo shoots are seasonal and have a short post-harvest shelf life, they must be properly processed for preservation and transport. Some species also require specific preparation methods to reduce bitterness and eliminate anti-nutritional factors. In India and other Southeast Asian countries, traditional preparation methods include washing, boiling, soaking, sun drying, and fermentation [81]. Today, advanced techniques are widely used in countries such as China, Taiwan, Korea, and Thailand. These include freeze drying, hot air drying, oven drying, solar drying, osmotic dehydration, and canning [81,85,86].

Bamboo shoots’ short post-harvest shelf life and seasonality pose serious obstacles to their widespread use as a food source worldwide. Fresh shoots are usually only available for a short time each year, and if they are not processed promptly, they quickly deteriorate because of enzymatic activity and microbial spoiling [87,88]. Their worldwide accessibility is limited by their perishability, which limits their distribution beyond local or regional markets. However, shelf life and marketability can be significantly increased by combining modern technologies like freeze drying, canning, and osmotic dehydration with more conventional preservation techniques like fermentation, drying, and boiling [87,89,90,91]. Therefore, improvements in processing and preservation methods hold promise for getting around seasonality and short shelf life, which are obstacles, and making bamboo shoots a sustainable global food source.

In addition to the challenges of seasonality and perishability, concerns are expressed regarding the potential for overuse of bamboo if the demand for food and medicine increases dramatically on a global basis [92,93]. Bamboo grows quickly and is renewable, but unsustainable harvesting practices can endanger natural regeneration and reduce supply for ecological and local community needs, particularly when it comes to young shoots during times of high demand. Overharvesting may also affect biodiversity because bamboo is a primary or seasonal food source for many animal species, such as golden lemurs, giant pandas, and mountain gorillas. Reducing these risks requires the use of sustainable management practices, such as rotational harvesting, community-based conservation, and the incorporation of farmed bamboo species into agroforestry systems [94]. Together with better preservation techniques, these steps can guarantee that bamboo’s nutritional and therapeutic potential is fulfilled without endangering the long-term supply or ecological stability.

### 3.2. Nutritional Status of Bamboo

Fresh bamboo stalks are a good source of thiamine, niacin, vitamin A, vitamin B6, and vitamin E. In addition to minor amounts of phosphorus and selenium, they contain essential minerals such as potassium, sodium, calcium, magnesium, and iron. Notably, bamboo shoots have the highest iron (Fe) content among commonly consumed vegetables, such as carrots, spinach, amaranth, pumpkin, and cucumber, making them especially beneficial for children and pregnant women. Bamboo shoots contain 17 amino acids, eight of which are essential for human health [95]. They are particularly rich in tyrosine, which constitutes approximately 57–67% of their total amino acid content. Protein content in fresh bamboo shoots ranges from 1.49 g/100 g to 4.04 g/100 g [81]. Bamboo shoots are also high in carbohydrates. For example, the edible shoots of *B. nutans*, *B. vulgaris*, *D. strictus*, and *D. asper* contain 3.3%, 3.4%, 0.6%, and 2.9% carbohydrate, respectively [96]. The high cellulose content in bamboo shoots aids digestion by enhancing peristaltic movements in the intestines [97]. Their low fat and high fiber content make them a desirable food for individuals managing modern lifestyle diseases [96]. Both fresh and fermented bamboo shoots are known to contain substantial amounts of phytosterols. Additionally, bamboo leaves exhibit strong antioxidant properties due to their high phenolic content. In the shoots of *P. pubescens*, eight phenolic acids have been identified, with protocatechuic acid, p-hydroxybenzoic acid, and syringic acid being the most abundant [98,99].

### 3.3. Bamboo in Pharmaceuticals

Bamboo has long been revered for its significant medicinal properties. Its use in pharmaceuticals underscores its potential to promote health and treat a variety of ailments. Traditionally, bamboo extracts have been utilized in specific cultural contexts due to their perceived therapeutic effects. Bamboo shoots and leaves are rich in essential nutrients and bioactive compounds, including vitamins, minerals, amino acids, and antioxidants. These are commonly extracted and incorporated into dietary supplements to support overall health and well-being. Bamboo extract supplements are known to aid in detoxification, improve digestion, and boost immune function [100,101]. Bamboo leaves, in particular, are rich in flavonoids, phenolic acids, and other antioxidants that combat oxidative stress, helping reduce the risk of chronic conditions such as cancer, cardiovascular disease, and neurodegenerative disorders [102]. Pharmaceutical products incorporating bamboo extracts can harness these antioxidant effects to promote cellular health and longevity.

Bamboo extracts also exhibit notable anti-inflammatory and antimicrobial properties [103], making them suitable ingredients in formulations targeting infections and inflammatory conditions. Topical products, such as creams and ointments, based on bamboo extracts are used to treat skin infections, wounds, eczema, and psoriasis. Silica, a major structural component of bamboo, plays a critical role in bone health [12]. Bamboo extracts are included in treatments aimed at preventing osteoporosis and improving bone density, particularly in postmenopausal women [104]. Bamboo shoots, being rich in dietary fiber, further support digestive health and are used in the management of gastrointestinal disorders, including irritable bowel syndrome (IBS) [80]. In recent years, the potential anticancer properties of bamboo extracts have attracted growing scientific interest. Various studies have indicated that bamboo may exert beneficial effects against cancer cells [105]. For example, a dietary supplementation of 0.5% with an ethanol/water extract of *P. edulis* significantly inhibited mammary tumor development in rats exposed to the carcinogen DMBA [106]. The extract reduced tumor incidence by 44% and decreased tumor multiplicity. Mechanistically, it downregulated the expression of estrogen receptors (ER-α and ER-β), HER2, and glutathione-S-transferase (GST) isoforms, suggesting its capacity to inhibit breast cancer progression and reduce chemoresistance.

The therapeutic potential of bamboo extract in cancer treatment is promising. Future research should aim to isolate and characterize specific bioactive compounds to better understand their mechanisms of action. Clinical trials are also necessary to evaluate safety, efficacy, and appropriate dosing in humans. Bamboo extracts could potentially be developed as adjuvant therapies, enhancing the effectiveness of conventional cancer treatments and reducing resistance. As evidence continues to accumulate, bamboo may emerge as a valuable natural agent in the fight against various forms of cancer.

## 4. Bamboo as a Source of Cosmetics

Bamboo extracts are increasingly used in skincare products due to their high silica content, amino acid composition, and antioxidant properties [107]. For example, silica, the most common constituent of bamboo leaves, mainly varies from 75.90 to 82.86%, stimulates the production of collagen and makes the skin firmer [108]. According to the species and extraction technique, the phenolic content varies from 2.5 to 8.7 mg GAE/g dry extract, and the corresponding antioxidant activities are assessed using DPPH radical scavenging assays at 55–78% inhibition [109]. Methionine, arginine, and threonine are among the amino acids that have been measured at 3–12 mg/g dry weight [109,110,111]. These quantitative values support the importance of bamboo-derived chemicals in cosmetic applications. These bioactive constituents are known to nourish, soothe, and rejuvenate the skin. For instance, the high silica content of bamboo extracts serves a similar purpose as well-known mineral-based substances like silica powders, which encourage the production of collagen and make skin firmer [112,113]. Green tea polyphenols and vitamin E, two well-known botanical antioxidants that are well-known for reducing oxidative stress in skin care products, have antioxidant properties that are similar to those of bamboo extracts [114]. Bamboo extracts are incorporated into a variety of cosmetic products, including moisturizers, serums, and facial masks. They are believed to improve skin elasticity, reduce the appearance of wrinkles, and promote overall skin health [115]. Bamboo charcoal is widely utilized in cleansing formulations such as facial cleansers, scrubs, and masks, owing to its excellent absorptive properties [112]. It has the ability to draw out impurities, excess oils, and toxins from the skin, facilitating deep and effective cleansing [116]. Additionally, bamboo fibers are gaining popularity in the production of cosmetic applicators like makeup brushes and sponges. Their soft and hypoallergenic nature makes them a viable, eco-friendly alternative to synthetic materials, offering a more natural and sustainable option for cosmetic application [117]. Another growing trend is the use of bamboo-based packaging in the cosmetics industry as a sustainable substitute for traditional plastic packaging. Bamboo is biodegradable, renewable, and has a lower environmental footprint than plastics [16,17]. Companies are increasingly adopting bamboo to manufacture packaging containers, caps, and other cosmetic product components [118]. Bamboo is also valued for its light, refreshing natural aroma, which is used in the formulation of perfumes and cosmetic products. These bamboo-derived fragrances evoke a sense of natural purity and relaxation, appealing to consumers seeking wellness-oriented and nature-inspired scents.

### Bamboo in Fabrics and Bio-Composites

Bamboo fibers are increasingly utilized in the textile industry due to their softness, breathability, antibacterial properties, and hypoallergenic nature. A primary motivation for using bamboo in textiles is its environmental sustainability, helping reduce the ecological footprint of the fashion industry. Viscose bamboo textiles have been developed for the footwear industry, exhibiting desirable mechanical properties such as high tear resistance, tensile strength, and elongation at break. These characteristics ensure that bamboo fabrics can be efficiently incorporated into footwear without showing signs of abrasion under functional testing [119]. Bamboo fabrics are also ideal for sportswear and activewear due to their inherent moisture-wicking and antibacterial properties, which enhance comfort and performance during physical activity [120]. In home textiles and furnishings, bamboo-based products include bed linens, towels, curtains, and upholstery. These offer a natural, sustainable alternative to conventional materials, combining comfort with ecological responsibility. Recent advancements in textile technology have enabled bamboo fibers to be engineered with specialized properties such as UV resistance and fire retardancy, expanding their use in technical textiles for niche applications [121].

In addition to fabrics, bamboo is widely used in the production of bio-composites. Bamboo-reinforced polymer composites are increasingly applied in the construction industry for the fabrication of panels, boards, and structural components. These bamboo-based materials present a sustainable alternative to conventional construction products [122]. Bamboo fibers combined with biopolymers are also used in packaging, offering a biodegradable, bio-based alternative to conventional plastic packaging [16,17]. Bamboo bio-composites are finding widespread use in consumer products as well, offering an ideal blend of aesthetic appeal and sustainability. These materials are used in furniture, home accessories, and casings for electronic devices [123]. Bamboo is also a promising precursor for activated carbon production. Through carbonization and activation processes, bamboo is transformed into highly porous activated carbon used in water purification, air filtration, and energy storage applications. Furthermore, bamboo-plastic composites are developed by embedding bamboo fibers into plastic matrices, resulting in materials that are biodegradable, lightweight, and suitable for diverse industrial uses [16,17,124]. Bamboo fibers or strips are also used as reinforcement in concrete, significantly enhancing its tensile strength and resistance to cracking [125].

Bamboo textiles and bio-composites are commonly promoted as sustainable alternatives, but their environmental benefits depend on the entire product life cycle [126]. Bamboo is renewable and requires little input to grow, but processing methods like making viscose can be very energy and chemical intensive. Life cycle assessments show that bamboo products frequently have lower carbon footprints and greater biodegradability than synthetic fibres and plastics, though results vary depending on how they are processed, transported, and disposed of. Cleaner technology and effective end-of-life management are essential if bamboo-based products are to have a significant positive environmental impact [12,127]. Although bamboo is a resource that grows quickly and is renewable, its actual impact on reducing emissions and promoting the green transition will depend on how the product is grown, processed, transported, and managed at the end of its useful life. To measure the net environmental benefits and prevent overestimating their impact, life cycle assessments are crucial.

## 5. Application of Bamboo in Biotechnology

Bamboo has emerged as a valuable plant matrix in the fields of biotechnology and bioprocessing. It can be used to produce sugars, prebiotic ingredients, sweeteners, and other compounds relevant to the food and pharmaceutical industries. Additionally, bamboo is highly suitable for biorefinery applications, primarily focused on the production of biofuels and industrial biochemicals [128]. While researchers have traditionally focused on lignocellulosic feedstocks such as sugarcane bagasse and straw [129], as well as rice and wheat straw [130], recent studies have shifted attention toward alternative biomasses with higher cellulose and hemicellulose content [131]. In this context, bamboo stands out due to its significant biotechnological potential. Characterization studies have shown that certain bamboo species, such as *D. latiflorus* and *Guadua amplexifolia*, possess high cellulosic (35–45%) and (20–30%) hemicellulosic fractions, making them competitive with other commonly used biomass sources in biotechnology. Compared to other lignocellulosic materials, bamboo offers several advantages: high productivity and biomass density, lower production and transportation costs, minimal fertilizer requirements, and propagation through clonal methods that eliminate the need for seed cultivation. These traits reduce cultivation costs and enhance culm and shoot production [117].

Beyond its role as a lignocellulosic feedstock, bamboo is increasingly recognized as a versatile platform across different domains of biotechnology (Figure 4) [132]. Tissue culture for large-scale propagation, genetic engineering/CRISPR for targeted trait improvement, and bioprocessing for turning biomass into fuels, chemicals, and high-value products are all applications of industrial biotechnology. To speed up breeding and innovation, forest biotechnology uses state-of-the-art techniques like transcriptomics, artificial intelligence, and genome and epigenome studies to identify the molecular mechanisms underlying bamboo growth, stress tolerance, and adaptation. Bamboo is used in agri-environmental biotechnology to produce biochar, which improves soil fertility, sequesters carbon, and promotes environmental sustainability, as well as in phytoremediation, which removes pollutants and heavy metals from soils. When taken as a whole, these uses highlight bamboo’s potential as a feedstock for biorefineries as well as its versatility as a resource for forestry, industrial, and environmental biotechnology.

The hemicellulosic fraction of bamboo contains xylan, which can be hydrolyzed into xylose, a pentose sugar used in ethanol production, and can also be further processed into high-value-added products such as xylo-oligosaccharides (XOS, prebiotics with food and pharmaceutical applications) and xylitol (a commercial low-calorie sweetener) [133]. Likewise, the cellulosic fraction yields glucose, a hexose sugar widely used as a substrate in alcoholic fermentation. Bamboo can also serve as a source of cell-oligosaccharides, which are of growing interest in functional food and nutraceutical industries [117].

Many bioprocess engineering methods, including self-hydrolysis, enzymatic hydrolysis, and chemical hydrolysis (either acidic or alkaline), have been investigated to create these products [134]. High rates of cellulose and xylan breakdown are produced by chemical hydrolysis, especially when high pressures and temperatures (120–180 °C) are applied, or when acidic/alkaline agents (0.5–2 M H_2_SO_4_ or NaOH) are utilised. There have been reports of 80–92% monosaccharide yields for glucose and 65–78% for xylose; however, it can also produce fermentation inhibitors such as hydroxymethylfurfural (HMF) and furfural [135]. However, while being more costly, enzymatic hydrolysis is advantageous due to its gentle operating conditions (30–50 °C, pH 4.5–5.5) and selectivity, which reduces inhibitory by-products.

To convert cellulose and hemicellulose into fermentable sugars, enzymatic processes usually employ cellulases (endoglucanases, exoglucanases, β-glucosidases) and hemicellulases (xylanases, auxiliary enzymes including arabinofuranosidases and acetylxylan esterases). Depending on the bamboo species and pretreatment, total sugar concentrations can vary from 90 to 110 g/L. Under optimal enzyme loading (15–30 FPU/g cellulose) and hydrolysis time (48–120 h), conversion efficiencies of 70 to 85% for glucose and 60 to 75% for xylose have been reported [136]. These sugars then serve as substrates for fermentation. Saccharomyces cerevisiae has been used to produce 40–50 g/L of ethanol, which is equivalent to 85–90% of the theoretical maximal conversion under controlled conditions (30 °C, pH 5.0, 150 rpm, 48–72 h). Engineered Escherichia coli bacteria have been reported to convert xylose-rich fractions into ethanol and xylitol with a yield of 35–40 g/L xylitol and a conversion efficiency of 70–75% [136]. As with other lignocellulosic feedstocks, distillation and purifying processes are typically part of downstream processing. These quantitative results demonstrate bamboo’s promise as a high-yield, sustainable source of chemicals, biofuels, and nutraceutical components, despite the fact that it has not yet been extensively employed on an industrial scale [137].

## 6. Bamboo as a Structural Material

Bamboo is a highly adaptable material with broad applications in construction. Due to its architectural versatility, the bamboo culm can serve as a direct structural element or be processed into various engineered forms. It offers socioeconomic benefits, being both durable and cost-effective [138]. Bamboo can be integrated into organic architectural designs and serves as a viable alternative to conventional construction materials [139]. Remarkably, bamboo possesses ultimate tensile strength comparable to that of steel. Bamboo’s tensile strength is reported at approximately 28,000 psi (pounds per square inch), slightly exceeding that of mild steel, which has a tensile strength of about 23,000 psi [140]. Additionally, bamboo fibers are longer than wood fibers, providing several technological advantages, including enhanced stiffness, durability, and superior physical and mechanical properties. Bamboo also exhibits a significantly higher strength-to-weight ratio than wood [125].

Furthermore, its strength and flexibility make bamboo well-suited for use in shelters designed to withstand earthquakes and extreme weather conditions. It can endure high-velocity winds and seismic activity measuring between 7 and 9 on the Richter scale. The lightness and flexibility of bamboo also make it ideal for prefabricated housing. Its processing consumes less energy compared to traditional wood materials. Bamboo offers acoustic insulation and ultraviolet protection, while its straightforward tools, techniques, and assembly methods make it accessible even for individuals with minimal construction experience. The following sections discuss the primary methods of using bamboo in construction.

### 6.1. Use of Full/Half Culm Bamboo in Construction

In regions where bamboo is abundant, full or half culms are frequently used in construction. This traditional application includes uses in resorts, scaffolding, columns, wall panels, roof purlins, and poles. Bamboo culms, whether whole or halved, are fastened together for various structural applications. However, due to the hollow nature of bamboo culms, making strong and reliable connections can be challenging. A typical failure mode for bamboo loaded parallel to the grain involves splitting and end-bearing failure, with the latter being exacerbated by high moisture content. Traditional joining techniques often involve tying culms together with rope, but modern methods, such as hose clamps, concrete infill, and bolted connections, have significantly improved the mechanical performance of bamboo structures [141].

### 6.2. Use of Engineered Bamboo in Construction

Engineered bamboo refers to bamboo that has been processed and transformed into standardized forms for enhanced structural performance. It is used in a variety of construction applications including flooring, wall sheeting, panels, and framing elements. Among the most common forms of engineered bamboo is laminated bamboo. To produce laminated bamboo, culms are first crushed and flattened into uniform strips, which are then glued together in layers to form boards [142,143]. These laminated boards behave orthotropically, exhibiting much greater compressive strength along the grain direction compared to the transverse directions [144]. Laminated bamboo is often used as a substitute for timber plywood, although its properties differ significantly. Another advanced product is bamboo scrimber, which is made by crushing and rolling bamboo strips into fiber bundles. These bundles are impregnated with adhesives and then hot-pressed to form dense structural boards. Commercial bamboo scrimber typically has a density ranging from 800 to 1200 kg/m^3^. Both its mechanical properties and water absorption characteristics are influenced by its density [145]. Studies have shown that bamboo scrimber and laminated bamboo outperform many conventional timber products in mechanical strength and durability [146]. Other engineered bamboo products include veneers, fire-resistant boards, mat boards, bamboo corrugated sheets, and veneer composites, all of which present viable, sustainable alternatives to traditional wood-based construction materials [147].

## 7. Use of Bamboo in the Manufacturing Industry

The bamboo manufacturing sector encompasses a wide range of sub-industries, including the production of bamboo fibers, household items, and handcrafted products. High-quality natural bamboo is used to produce bamboo fibers through a high-tech extraction process that isolates cellulose fibers, which are then processed into various materials. Bamboo fibers find application in construction, papermaking, and the textile industry [148]. Bamboo also exhibits excellent gas permeability, high water absorption, strong abrasion resistance, and dyeability, resulting in visually appealing products [149]. Additionally, bamboo has natural antibacterial and insect-repellent properties and emits negative ions, making it highly suitable for a wide range of industrial applications. Products made from bamboo fiber often command premium prices due to these functional benefits. Although the current market share for bamboo-based products remains relatively small, it is expected to grow significantly in response to stricter environmental regulations and rising living standards.

This anticipated growth may lead to supply shortages in raw bamboo materials. Therefore, bamboo-producing regions, particularly in China, must scale up raw material production by establishing regulated bamboo plantations and diversifying their sources of bamboo fiber. As eco-friendly, non-toxic household goods become more popular with technological and societal advancement, there is a need to expand product offerings to include bamboo-based items such as guardrails, doors, windows, handrails, and stairs. In parallel, efforts must be made to improve the production of bamboo-laminated timber and composite furniture. Developing strong household bamboo brands is also critical to improving product design, structural functionality, and overall competitiveness [150,151]. A key component of the bamboo manufacturing industry is the production of bamboo crafts, traditionally created using bamboo materials [152]. To meet rising demand, it is essential to increase production volumes, improve the diversity and quality of bamboo craft products, and promote greater utilization of bamboo as a raw material [153]. This includes the coordination of production for items like bamboo fans and the preservation and innovation of traditional crafts such as bamboo root carving, cutting, and calligraphy.

Modern bamboo crafts now include a variety of products such as bamboo paintings, carpets, pendants, jewelry boxes, and tobacco packaging boxes. These items emphasize artistic and aesthetic appeal. New innovations are also emerging in functional bamboo products, including keyboards, computer casings, and automotive interior panels. Bamboo fiber has proven useful in both the textile and automotive industries. For example, bamboo fiber can be used to produce socks and towels. These creative applications not only enhance tourism through local craft markets but also foster economic development in bamboo-growing regions [154]. Bamboo can be sold as raw material or transformed into high-value crafts and fiber products, often fetching better prices.

As highlighted in Section 3.2 and Section 3.3, bamboo shoots offer significant health benefits and possess strong market potential. The global market for both fresh and dried bamboo shoots is expanding rapidly, yet supply still lags behind demand [155]. Even the world’s largest producers struggle to satisfy both domestic and international markets. Well–managed bamboo plantations produce significantly more shoots, generate higher returns than traditional grain crops, and incur lower management costs. Establishing bamboo plantations can deliver immediate economic benefits [156]. Bamboo and its shoots offer dual–income opportunities: they can be sold directly as edible raw materials or processed into high–value food products. Traditional consumption of bamboo shoots spans many cultures worldwide [157], and modern industrial methods now enable the production of commercial items such as bamboo vinegar and juice-based beverages. In particular, flavonoids extracted from bamboo leaves serve as key ingredients in bamboo juice drinks, further enhancing the market value of this versatile plant.

## 8. Use of Bamboo in Sustainable Agriculture and Agroforestry

Bamboo is increasingly utilized to restore degraded lands due to its ability to thrive in nutrient-poor soils [158]. It can grow across diverse environments, from regions exposed to strong winds to areas with intense sunlight, making it ideal for land rehabilitation. Bamboo species serve as pioneer plants in eco-restoration, owing to their adaptability and efficiency in conserving water and nutrients [159]. Their rapid growth and dense foliage promote the accumulation of a thick litter layer, which helps maintain soil moisture and regulate the understory microclimate, both critical for rehabilitating degraded landscapes. Bamboo’s extensive root and rhizome system contributes to effective soil stabilization. There are two primary rhizome types: monopodial (“running bamboo”), which spreads quickly through horizontal growth, and sympodial, which grows in compact clumps. Rhizome buds may develop horizontally to extend the rhizome network or vertically to form new culms. On degraded jhum lands and hilly slopes, these root systems help reduce runoff and control soil erosion. However, there are limited studies on the natural colonization of damaged lands by bamboo.

Certain bamboo species are also known to enhance soil fertility by increasing microbial biomass in the rhizosphere. Among 11 species investigated, *D. giganteus*, *D. hookerii*, and *B. nutans* were most effective in improving and maintaining the fertility of acidic soils in the Northeastern Himalayan region of India [160]. Microbial biomass plays a key role in nutrient cycling, particularly of nitrogen and phosphorus [161]. Bamboo acts as both a “source” and “sink” for soil nutrients. Compared to monocultures, mixed bamboo stands provide improved soil bulk density, aeration, porosity, and nutrient availability. Bamboo plantations have also been shown to enhance microbial activity, enzymatic function, and overall soil fertility [162]. Likewise, bamboo forests significantly improve soil physicochemical properties [163]. In coastal areas, planting bamboo on sandy soil has promoted the growth of tree species and strengthened coastal shelterbelt forests.

In Sub-Saharan Africa (SSA), where agriculture remains the backbone of many national economies, bamboo-based agroforestry offers a promising pathway for sustainable development. The design of agroforestry systems is influenced by the bamboo type, sympodial or monopodial, based on their morphological and physiological characteristics. Sympodial bamboos, suited to temperate climates with cool, moist winters, produce new culms from buds on elongated culm necks (pseudo-rhizomes). In contrast, monopodial bamboos, typically found in tropical regions with distinct dry seasons, sprout from rhizome-based buds [164].

Most cultivated bamboo species in SSA are sympodial and found in forest agroecological zones. These are particularly well-suited for fallow planting on degraded soils, integration into shelterbelt systems, use as windbreaks, and for boundary planting. India’s successful implementation of agroforestry systems with sympodial bamboo species offers a replicable model for SSA. Establishing similar systems using *B. multiplex*, *B. vulgaris* (green type), *B. bambos*, *B. pervariabilis*, *B. vulgaris* var. *vittata*, *Oxytenanthera abyssinica*, and *D. strictus* could provide a sustainable biomass base for firewood and charcoal production. Bamboo-based windbreaks and boundary planting can reduce crop failure risks, establish clear farm borders, and mitigate land disputes. Furthermore, riparian buffer zones, particularly where natural stands of *B. vulgaris* occur, present unique opportunities for bamboo-based watershed protection, supporting both biodiversity and water quality [165].

Beyond sympodial types, monopodial bamboos also perform well in intercropping systems. Depending on the clump size, they can be planted at spacings ranging from 4 × 9 m to 8 × 9 m. Studies in India have identified crops such as finger millet, cowpea, bottle gourd, turmeric, sesame, and sweet potato as particularly compatible for intercropping with bamboo [166]. Given the high nutritional content of bamboo leaves, bamboo-based silvopastoral systems offer strong potential in SSA, where livestock farming is a primary livelihood. Bamboo can serve as a high-quality fodder source, especially during the dry season when other forages are scarce [167]. Even in systems where bamboo canopy cover inhibits food crop cultivation, its role in supplemental livestock feeding enhances the ecological and economic sustainability of bamboo-based agroforestry.

## 9. Biological, Agronomic, and Genetic Advances in Bamboo: Challenges, Stress Responses, and Prospects for Improvement

### 9.1. Physiological and Agronomic Characteristics of Bamboo Cultivation

Bamboo is a type of grass that grows rapidly and regenerates through its rhizome system after harvesting [12]. It has unique physiological and agronomic traits that make it different from other woody plants and perfect for sustainable farming. Its amazing growth rate, especially in species like *Phyllostachys edulis*, is due to high cell wall extensibility, auxin-regulated cell division, and a well-coordinated hormonal interaction between gibberellins (GA), cytokinins, and abscisic acid (ABA) [1]. Under ideal light and water circumstances, bamboo exhibits C3 photosynthesis more efficiently than other grasses, allowing for a rapid accumulation of biomass [168]. Furthermore, its vast rhizome-root system facilitates vegetative propagation and improves resource acquisition, enabling strong colonisation of marginal or degraded lands [169]. Agronomists say that bamboo grows best in places that receive between 1000 and 2500 mm of rain a year and have well-drained, loamy soils with a pH between 5.5 and 6.5 [170]. However, management techniques including mulching, rhizome barrier control, appropriate spacing, and thinning sometimes to lessen intra-clump competition are also necessary for its successful development [30]. Despite these favourable circumstances, output may drastically decline in areas susceptible to waterlogging, in inappropriate soils, or during drought stress. These restrictions necessitate site-specific management packages, which may increase costs and restrict smallholders’ access to the crop [171]. A major mechanism governing shoot emergence and propagation cycles, physiological dormancy in bamboo buds is governed by photoperiod and seasonal temperature fluctuations. It can be divided into two categories: endodormancy, which is internally regulated, and ecodormancy, which is environmentally regulated [172]. Because physiological reactions and outside environmental cues interact, agronomic packages that are specific to a certain area and bamboo species are needed. Also, bamboo is a strong resource for making biomass, planting trees again, and stopping soil erosion because it can quickly grow back after being cut down by sprouting new rhizomes [173]. Nevertheless, excessive rhizome proliferation may sometimes lead to invasiveness or competition with neighboring vegetation, posing ecological management challenges in mixed agroforestry systems [174]. To maximise productivity and ensure long-term viability in bamboo-based agroforestry systems, it is essential to comprehend not only the physiological traits that regulate bamboo’s growth and reproduction but also the management and environmental constraints that may prohibit its widespread use.

In addition, bamboo has a number of distinct biological traits that set it apart from other woody plants and grasses, especially in terms of its developmental biology and growth dynamics. Its monocarpic flowering habit is one of its most striking characteristics; many species, including *P. edulis*, flower simultaneously following decades of vegetative growth before experiencing senescence and death, a phenomenon associated with hormonal changes and programmed cell death (PCD) [175,176,177]. This widespread death poses a serious threat to the sustainability and economic return of plantations. The molecular basis of this trait includes the overexpression of genes like *SOC1*, *FT*, and *AP1* that regulate floral initiation as well as the downstream activation of genes associated with PCD, which accelerates the degradation of vascular tissues and meristems [178,179]. From an agronomic perspective, this unpredictable flowering behaviour demands prompt attention. Breeding or genome-editing interventions aimed at delaying flowering or creating non-flowering clones, early detection using floral biomarker gene expression, and spaced planting to avoid synchronous dieback are some methods to mitigate its effects [180]. Sustenance of long-term coordinating management of a plantation as well relies on ex situ conservation, flower cycle data recording across different areas and cryopreservation of elite lines. Therefore, an understanding of the biology of the growth and death cycles is necessary in bamboo for its successional productivity and sustainable utilization, as well as for ecological research.

Although most physiological and agronomic research focusses on commonly grown species like *P. edulis*, the genus Bambusoideae contains a wide variety of species, many of which have distinct growth patterns, flowering cycles, and resource needs [12,181]. Practical difficulties arise when cultivating and processing less common species because some have slower growth rates, irregular rhizome proliferation, or climatic and soil preferences that restrict their adaptability [181,182,183]. Furthermore, different culm structures, lignin contents, or fibre properties may require customised harvesting methods and post-harvest processing. Developing management plans, creating agronomic packages that are species-appropriate, and encouraging the sustainable use of the entire range of bamboo diversity all depend on an understanding of these species-specific variations [126].

### 9.2. Biotic Constraints: Pest and Diseases in Bamboo

Bamboo cultivation is seriously afflicted by a number of pests and diseases, resulting in low-yield, poor culm quality and endangering the sustainability of plantations, despite the resistance and good growth of the crop. The single most serious insect pest is the bamboo shoot borer (*Omphisa fuscidentalis*), which tunnels into young shoots and causes extensive injury throughout the growing season [184]. Other common insect pests causing defoliation, yellowing, and reduced photosynthetic efficiency of bamboos include aphids, mealybugs, scale insects, and the bamboo leaf roller [185,186]. Moreover, termites are a serious threat, especially to dried or stored bamboo culms [187]. Since infection with all these pests in their initial stages is usually unnoticed, integrated pest monitoring is necessary for timely identification and control.

Numerous fungal diseases can also harm bamboo, mostly affecting the leaves, culms, rhizomes, and roots [132]. Among the most common diseases are culm rot from *Fusarium* spp. and *Curvularia* spp., sooty mould from *Capnodium* species, and leaf blight from *Alternaria alternata*, particularly in areas with high humidity, dense plantations, and inadequate airflow [184,188,189]. Reduced culm strength, splitting, and discolouration are common outcomes of fungal infections, which eventually affect bamboo’s market value. Bamboo is so frequently marketed as a hardy crop, but its susceptibility to numerous pest and disease combinations emphasises the need for careful management and continuous study to ensure sustainable production.

The use of integrated pest and disease management (IPDM) techniques is growing in response to these biotic restrictions [190]. These include parasitoids for the suppression of insect pests and biological control agents like *Trichoderma harzianum* and *Beauveria bassiana* [191]. Furthermore, organic mulching and the application of botanical pesticides (such as neem extract) lower pest burdens and enhance soil microbial health [192]. The discovery of resistance gene analogues (RGAs) and defense-related transcription factors like WR [193]. KY and NAC have bolstered recent efforts to create disease-resistant clones using tissue culture and molecular breeding techniques [180,194]. However, successful mite management is obstructed by a lack of comprehensive pest surveillance information, an inadequate disease diagnosis infrastructure, and low farmer knowledge. To ensure the sustainability of bamboo plantation, there is an open call that future research should be focused on development of species specific IPDM practices, pest prediction models and easily accessible training to stakeholders.

### 9.3. Bamboo Genetics: Resistance Genes and Targets for Crop Improvement

New opportunities for crop improvement have been made possible by developments in bamboo genetics, especially when it comes to problems with disease resistance, stress tolerance, flowering management, and biomass output. *Phyllostachys edulis* genome sequencing has made it possible to identify a large number of genes linked to stress response and growth regulation [195]. These resistance gene analogues (RGAs), like the nucleotide-binding site leucine-rich repeat (NBS-LRR) gene, are essential for identifying pathogens and triggering immune responses [196]. The expression of transcription factors like WRKY, NAC, MYB is significantly enhanced under biotic and abiotic stress conditions that have been correlated with enhanced resistance towards drought, fungi and fungal infection. Genes related to lignin synthesis (e.g., *PAL*, *C4H*, and *COMT*) are also targets of genetic modification because they affect culm strength and disease resistance [197]. Simultaneously, genes including *SOC1*, *FT*, and *LFY* are crucial regulators of bamboo blooming timing, which is especially crucial for postponing monocarpic flowering events that interfere with farming cycles [178,198,199]. Attempts are underway to identify quantitative trait loci (QTLs) and develop molecular markers for traits such as resistance to pests, biomass production, and shoot emergence. Advances in omics technologies such as the transcriptomics, proteomics and metabolomics have facilitated gene identification and functional characterisation (especially driven by the transcriptome efforts) [180]. Furthermore, preliminary research on the possible use of CRISPR/Cas9 genome editing in bamboo is concentrating on genes linked to stress response, rhizome management, and plant architecture [180,200]. These genetic technologies have enormous potential in the development of elite bamboo varieties with improved resistance, optimized biomass and culm yield, and ecological sustainability, albeit with continued challenges in transformation efficiency and regulatory approval.

### 9.4. Progress in Sustainable Cultivation, Conservation and Utilization

Growing awareness of bamboo’s ecological, economic, and social advantages has led to notable advancements in its sustainable cultivation, conservation, and use in recent years. At the cultivation level, developments in nodal culture, macroproliferation, and micropropagation techniques have made it easier to produce elite bamboo clones on a wide scale with desired characteristics including high biomass, quick growth, and disease and insect resistance [201]. These propagation methods ensure genetic purity, reduce reliance on seeds, which are often unavailable, and promote year-round cultivation because bamboo only flowers once a year. Conservation efforts are being strengthened by the establishment of ex situ germplasm banks, in situ protected reserves, and cryopreservation of seeds and tissues for rare and endangered species [12]. Molecular methods such as DNA barcoding and genome sequencing have also aided in the identification of species and the development of conservation strategies. Bamboo has emerged as a key component of green circular economies due to its applications in construction, furniture, textiles, paper, bioenergy, and biodegradable composites. If it is to be widely adopted, however, issues like a lack of high-quality planting material, species adaptation variance, and underdeveloped value chains that keep smallholders from reaping the benefits of a fair market must be resolved. Bamboo has been incorporated into plans for climate-smart agriculture, land restoration, and the development of rural jobs in nations like China, India, Thailand, and Ethiopia [202].

Bamboo is a critical resource in climate change mitigation due to its notable ability to sequester carbon quickly. The development of sustainable bamboo value chains has benefited greatly from policies like China’s bamboo industrial clusters, India’s National Bamboo Mission, and international partnerships through INBAR [199,203] (https://drawdown.org/solutions/bamboo-production; accessed on 22 August 2025; https://www.myscheme.gov.in/schemes/nbm; accessed on 22 August 2025). However, there are still challenges in scaling up high-quality planting material, standardising practices, and ensuring equitable access to markets for smallholder growers. All things considered, integrating indigenous knowledge systems, technological advancements, and supportive legislative frameworks is necessary to fully realise bamboo’s potential in sustainable development.

#### Technological, Logistical, and Economic Limitations

Bamboo is often praised for its economic and environmental benefits, but these benefits will not materialise unless significant financial, technological, and logistical barriers are eliminated. Bamboo has obvious ecological and financial benefits, but widespread adoption is still fraught with difficulties. Technological constraints include the need for sophisticated pretreatment and conversion technologies for biofuels to handle high lignin content and heterogeneous fibre composition, a lack of effective harvesting mechanisation, and challenges managing culms of different diameters and densities [204,205]. Bulky culms that are expensive to transport and store, a lack of processing infrastructure in non-traditional growing areas, and the perishability of bamboo shoots that makes supply chain management more difficult are the logistical challenges facing bamboo cultivation [17]. Economically speaking, farmer incentives are diminished by high initial establishment costs, protracted wait times for returns, and volatile market prices. Bamboo’s stubborn lignocellulosic structure, in particular, necessitates energy-intensive pretreatments and enzymatic hydrolysis in the context of biofuels. Additionally, the lack of large-scale biorefineries that are optimised for bamboo biomass further reduces its commercial competitiveness when compared to other biofuel feedstocks. To fully realise bamboo’s potential as a sustainable bioresource, these logistical and technological limitations must be addressed through processing innovations, infrastructure investments, and supportive policy frameworks [206]. Another issue with bamboo-based systems is nutrient management, particularly when there is phosphorus deficiency. Recent developments show how tailored biochars might support bamboo-based methods for managing soil and water quality. One example is lanthanum hydroxide–coated sludge biochar (La-C-550), which has a phosphorus adsorption capacity of 76.4 mg-P/g and steady removal in actual effluents [207].

## 10. Conclusions

From a traditional multipurpose plant, bamboo has evolved into a strategic bioresource at the nexus of biotechnology, industry, and ecology. It is a key component of low-emission and circular bioeconomies due to its remarkable growth rate, ability to sequester carbon, and adaptability to a variety of agroecological zones. The biotechnological application outlined in this review from tissue culture, genome editing, and bioprocessing to uses in biochar, phytoremediation, and artificial intelligence emphasize how versatile bamboo is in promoting agri-environmental and industrial innovation.

However, the use of bamboo has not been without its difficulties. The ecological hazards of large-scale monocultures, mechanical variability among species, and the absence of standardised engineering and harvesting codes demand immediate scientific and policy attention. To guarantee that bamboo lives up to its potential as a regenerative material, the environmental impact of processing techniques must also be taken into consideration.

In order to fully realise bamboo’s potential, it will be essential to invest in green processing technologies, promote transdisciplinary research, and incorporate bamboo into international climate strategies and carbon-credit frameworks. Bamboo can advance from specialised uses to become a key resource for Industry 5.0, promoting resilient supply chains, sustainable urbanisation, and regenerative rural economies, provided these issues are methodically resolved.

## Figures and Tables

**Figure 1 plants-14-03019-f001:**
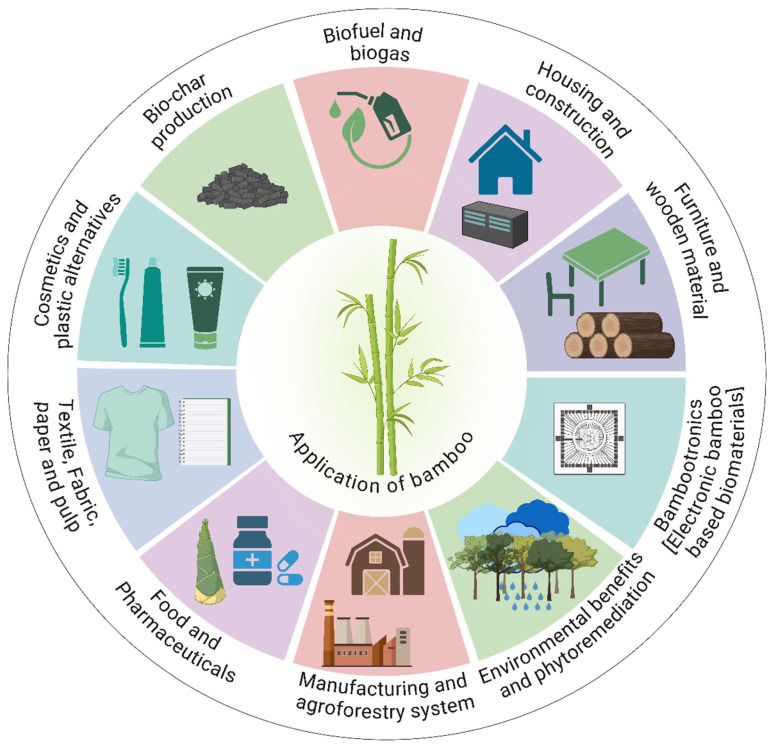
Traditional and emerging applications of bamboo across multiple industries. Bamboo serves as a multipurpose resource with applications spanning both traditional uses and modern technological advancements. Key sectors include housing and construction, furniture and wood products, food and biomass production, pharmaceuticals, environmental remediation, and cosmetics. Emerging innovations involve biochar production, biofuel and biogas generation, bamboo-based electronics (“bambootronics”), and the development of plastic alternatives [16,17]. Bamboo also plays a vital role in the textile, paper, pulp, manufacturing, and agroforestry sectors. Its exceptional versatility and sustainability position it as a cornerstone of future green economies and Industry 5.0 innovations.

**Figure 3 plants-14-03019-f003:**
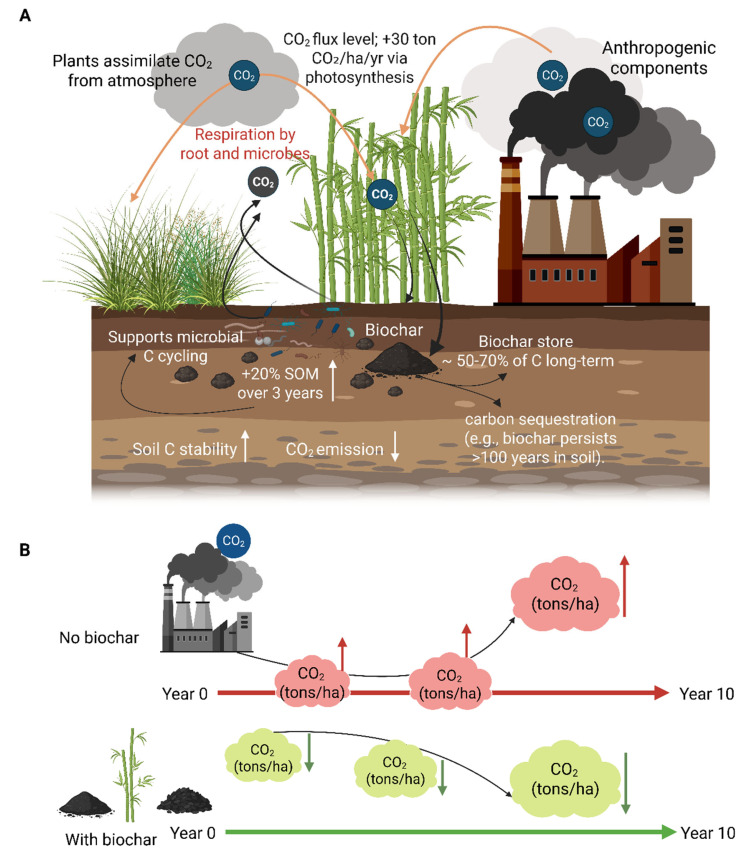
(**A**) This conceptual model illustrates the carbon sequestration potential of bamboo and biochar systems under industrial CO_2_ emissions. Biochar enhances soil organic matter (SOM) and stabilizes carbon, while plants assimilate atmospheric CO_2_ via photosynthesis. The figure integrates both natural (plant-microbe-soil) and anthropogenic (industrial emission) components of the carbon cycle. (**B**) Depicts the effect of bamboo biochar on atmospheric CO_2_.

**Figure 4 plants-14-03019-f004:**
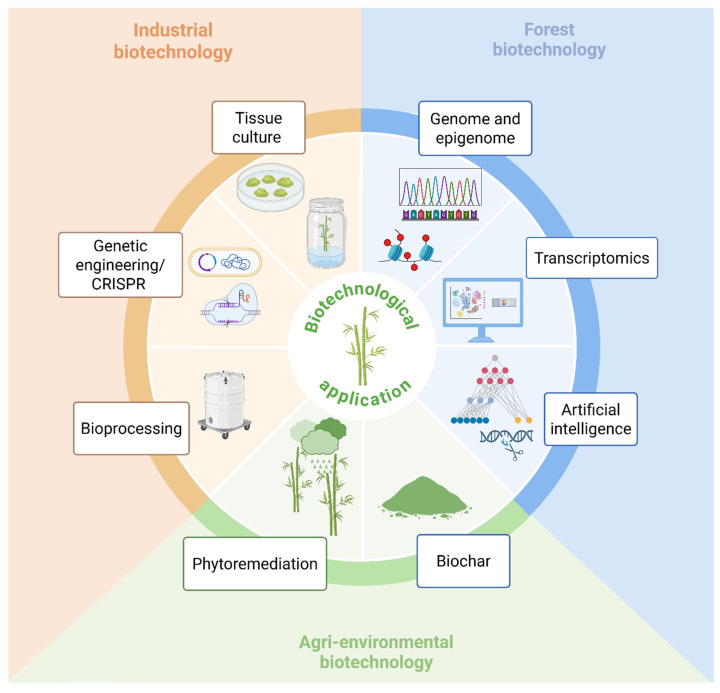
Biotechnological applications of bamboo. The figure illustrates the major domains of bamboo biotechnology categorized into three aspects: industrial biotechnology, forest biotechnology, and agri-environmental biotechnology.

**Table 1 plants-14-03019-t001:** Comparative mechanical and environmental properties of bamboo and conventional materials.

Property	Bamboo	Timber (Hardwood)	Steel	Plastic	References
Density (kg/m^3^)	600–800	500–900	~7850	900–970	[12,24]
Tensile strength (MPa)	140–370 (age & species dependent)	40–100	400–550 (mild steel)	20–40	[24,25]
Compressive Strength (MPa)	40–80	30–70	250–500	5–20	[26,27,28]
Elastic Modulus (GPa)	10–30	8–14	~200	0.8–1.5	[25,29]
Carbon Sequestration (t C/ha)	Up to 259	50–150	None	None	[30,31,32]
Renewability	Fast (3–5 years)	Slow (10–30 years)	Non-renewable	Non-renewable	[33,34]
Biodegradability	Biodegradable (untreated)	Biodegradable	Non-biodegradable	Non-biodegradable	[35,36]
Recyclability	Emerging (e.g., biochar, composites)	Limited	High	Limited	[35,37]
Flammability	High (untreated)	Moderate (moisture-dependent)	Non-flammable	Highly flammable	[36,38]
Cost (USD/ton)	80–120 (species/region dependent)	150–250	500–1000	1000–1500	[39]

## Data Availability

No new data were generated in association with this article.

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
