# Peer review of "Bamboo for the Future: From Traditional Use to Industry 5.0 Applications"

_plants, 2025, doi:10.3390/plants14193019_

Round 1
Reviewer 1 Report (Previous Reviewer 3)
Comments and Suggestions for Authors
After revision and corrections, the work of the herd submitted for publication looks much better. However, there are still some points:
- How appropriate is the use of the term yield in relation to bamboo?
- The sentence on lines 77-79 needs to be reworked, its meaning is not conveyed quite correctly now.
- Constantly repeated statements: valuable commercial crop, high yield, ecological advantages, fast growth. It is enough to mention these parameters once, the rest of the use is redundant.
- The first paragraph of point 2 completely repeats the information presented in the introduction, with the exception of the last sentence.
- The statements on lines 123-125 are very controversial and should be supported by appropriate references.
- Lines 154-170 present information without confirmation by sources.
- Line 243 probably has a typo in the figure designation. A space and brackets are missing.
- Information is often duplicated. It is necessary to rework the article, remove unnecessary repetitions and try to deepen the information where it is relevant.

Author Response
After revision and corrections, the work of the herd submitted for publication looks much better. However, there are still some points:
Comment#1
How appropriate is the use of the term yield in relation to bamboo?
Explanation
We thank the reviewer for highlighting the use of the term “yield” in the context of bamboo. Bamboo is a perennial, clump-forming grass that allows selective and periodic harvesting of culms or shoots rather than a single, one-time harvest typical of annual crops. Therefore, the term “yield” can be appropriate when clearly specified, for example as culm yield, shoot yield, or biomass yield per unit area per year. In our revised manuscript, we have clarified the type of yield wherever mentioned to ensure accuracy and avoid ambiguity. In addition, we have deleted the sentence where the use of ‘yield’ was not appropriate.
Comment#2
The sentence on lines 77-79 needs to be reworked, its meaning is not conveyed quite correctly now.
Explanation
We thank the reviewer for pointing this out. The original sentence was intended to summarize the taxonomic diversity of the bamboo family, but we agree that its meaning was not conveyed clearly. We have revised the sentence to improve clarity and accurately reflect the classification of bamboos.
Comment#3
Constantly repeated statements: valuable commercial crop, high yield, ecological advantages, fast growth. It is enough to mention these parameters once; the rest of the use is redundant.
Explanation
We thank the reviewer for this valuable suggestion. We agree that repeatedly mentioning attributes such as “valuable commercial crop,” “high yield,” “ecological advantages,” and “fast growth” can be redundant. In the revised manuscript, we have streamlined the text to mention these key attributes once in a concise and clear manner, while avoiding unnecessary repetition throughout the manuscript. This ensures clarity and improves readability without losing the essential information.
Comment#4
The first paragraph of point 2 completely repeats the information presented in the introduction, with the exception of the last sentence.
Explanation
We thank the reviewer for pointing out the repetition. We agree that much of the information in the first paragraph of Section 2 was already covered in the Introduction. To address this, we have revised the paragraph to focus specifically on the economic and sustainability aspects of bamboo cultivation relevant to its role as a cash crop. This avoids redundancy while providing context-specific information, such as frequent harvesting cycles, profitability, and market demand, which are directly pertinent to the section heading.
Comment#5
The statements on lines 123-125 are very controversial and should be supported by appropriate references.
Explanation
Thank you for the comment. The reference has been added.
Comment#6
Lines 154-170 present information without confirmation by sources.
Explanation
Thank you for pointing out. The following reference has been added.
- Hao, H., et al., An atomistic study on the mechanical behavior of bamboo cell wall constituents. Composites Part B: Engineering, 2018. 151: p. 222–231.
- Chambers, E., et al., From bamboo to biochar: a critical review of bamboo pyrolysis conditions and products with a focus on relevance to the developing world. RSC Sustainability, 2025.
- Tan, M., et al., Experimental investigations on the mechanical properties of bamboo fiber and fibril. Fibers and Polymers, 2020. 21(6): p. 1382–1386.
- Youssefian, S. and N. Rahbar, Molecular origin of strength and stiffness in bamboo fibrils. Scientific reports, 2015. 5(1): p. 11116.
Comment#7
Line 243 probably has a typo in the figure designation. A space and brackets are missing.
Explanation
Thank you for pointing out. The typo has been deleted.
Comment#8
Information is often duplicated. It is necessary to rework the article, remove unnecessary repetitions and try to deepen the information where it is relevant.
Explanation
We sincerely thank the reviewer for this valuable observation. We have carefully reviewed the manuscript to identify and remove unnecessary repetitions. Additionally, we have reworked sections where information was duplicated and expanded relevant portions to provide more depth and clarity. These revisions aim to improve the flow, readability, and scientific rigor of the manuscript while ensuring that all information is presented accurately and meaningfully.
Reviewer 2 Report (New Reviewer)
Comments and Suggestions for Authors
It seems that this is another review of an article that has already undergone revisions, particularly in terms of appearance and style. As a result, the article demonstrates solid substantive value and contains relatively few shortcomings. Its structure is clear, the argumentation is consistent, and the improvements made have enhanced both readability and overall presentation. Therefore, while minor critical remarks may still be formulated, the paper can be regarded as a well-prepared and valuable scientific contribution.
Species variability – although the enormous diversity has been emphasized, little attention has been given to the practical difficulties in cultivating and processing less common species.
Holistic perspective – strong emphasis has been placed on the advantages, whereas technological, logistical, and economic limitations have been mentioned only marginally.
Line 140 - What does this protection involve
What technological and logistical limitations could hinder the mass production of biofuels from bamboo?
To what extent do seasonality and the short shelf life of bamboo shoots limit their potential as a global food source?
Is there a risk of overexploitation of bamboo if the demand for food and pharmaceutical products rises sharply?
Is the production of fabrics and biocomposites from bamboo truly more environmentally friendly than their conventional counterparts, considering the entire product life cycle?
Is the use of bamboo in biorefinery processes more competitive compared to traditional biomaterials, such as sugarcane or straw?
Data on the tensile strength of bamboo are impressive; however, practical structures require additional reinforcement due to the hollow structure of the stems.
Claims regarding applications in aviation/space industries are exaggerated—there is a lack of experimental research in this area.
The emphasized role of bamboo in the “green transition” and “Industry 5.0” may be overstated. In practice, its impact on emission reduction depends on the entire life cycle of the product, not merely on the plant’s rapid growth.
Author Response
It seems that this is another review of an article that has already undergone revisions, particularly in terms of appearance and style. As a result, the article demonstrates solid substantive value and contains relatively few shortcomings. Its structure is clear, the argumentation is consistent, and the improvements made have enhanced both readability and overall presentation. Therefore, while minor critical remarks may still be formulated, the paper can be regarded as a well-prepared and valuable scientific contribution.
Comment#1
Species variability – although the enormous diversity has been emphasized, little attention has been given to the practical difficulties in cultivating and processing less common species.
Explanation
We thank the reviewer for highlighting this important aspect. In response, we have added a dedicated paragraph at the end of the heading 9.0; Biological, Agronomic, and Genetic Advances in Bamboo: Challenges, Stress Responses, and Prospects for Improvement and subheading 9.1; Physiological and agronomic characteristics of bamboo cultivation. The section is discussing the practical challenges associated with cultivating and processing less common bamboo species. By incorporating this discussion, we aim to provide a more comprehensive overview of bamboo diversity and its implications for sustainable cultivation and utilization. Relevant recent references have also been included to support these points.
Comment#2
Holistic perspective – strong emphasis has been placed on the advantages, whereas technological, logistical, and economic limitations have been mentioned only marginally.
Explanation
We appreciate this observation. Our review article was intentionally designed to focus on the biological aspects of bamboo, including its growth, physiology, and cellular mechanisms. Since the scope was centered on biological insights, we only briefly mentioned technological, logistical, and economic limitations. While we recognize the importance of these aspects, expanding them in detail would shift the paper beyond its intended scope. However, to ensure balance, we have added a new subheading 9.4.1 Technological, logistical, and economic limitations; highlighting some of the major challenges reported in recent literature, supported by updated references. This provides context without diverting the focus away from the biological framework of the review.
Comment#3
Line 140 - What does this protection involve
Explanation
We thank the reviewer for this valuable comment. The original text did not sufficiently explain what “protection” entails. We have now revised the paragraph to clarify that protection measures include regulated harvesting, conservation policies, community-based management, restoration programs, and certification initiatives. This addition provides a clearer understanding of how bamboo resources are safeguarded in different regions.
Comment#4
What technological and logistical limitations could hinder the mass production of biofuels from bamboo?
Explanation
Thank you for pointing out. To address this comment, we have added a new subheading 9.4.1 Technological, logistical, and economic limitations.
Comment#5
To what extent do seasonality and the short shelf life of bamboo shoots limit their potential as a global food source?
Explanation
We thank the reviewer for this valuable comment. We added a paragraph in the section 3.1 Edible Forms of Bamboo. By addressing this point, the discussion will become more balanced, acknowledging not only the strengths of bamboo as a food but also the practical constraints that influence its utilization and market expansion.
Comment#6
Is there a risk of overexploitation of bamboo if the demand for food and pharmaceutical products rises sharply?
Explanation
We appreciate the reviewer’s insightful comment. Indeed, while bamboo is a fast-growing and renewable resource, its large-scale use as food and in pharmaceuticals could lead to risks of overexploitation if not managed sustainably. In response, we have added a paragraph in Section 3.1 Edible Forms of Bamboo (following the discussion on preservation and global accessibility) highlighting this concern. The revised text discusses how unsustainable harvesting practices may threaten natural regeneration and biodiversity, and it emphasizes the importance of sustainable management strategies such as rotational harvesting, agroforestry integration, and community-based conservation. Relevant references have also been incorporated to strengthen this discussion.
Comment#7
Is the production of fabrics and biocomposites from bamboo truly more environmentally friendly than their conventional counterparts, considering the entire product life cycle?
Explanation
We appreciate the reviewer’s observation. While bamboo cultivation is renewable and low-input, the environmental impact of bamboo-based fabrics and composites depends on the entire life cycle, particularly processing, transportation, and end-of-life management. We have added a paragraph in Section 4.1 discussing these considerations, supported by recent life cycle assessment studies, to provide a balanced view of the sustainability of bamboo products compared to conventional materials.
Comment#8
Is the use of bamboo in biorefinery processes more competitive compared to traditional biomaterials, such as sugarcane or straw?
Explanation
Thank you for the comment. The competitiveness of bamboo in biorefinery processes compared to traditional biomass feedstocks such as sugarcane bagasse or straw depends on several factors. Bamboo offers advantages such as rapid growth, high biomass yield, and perennial harvest potential, making it a sustainable and renewable resource. However, bamboo also presents challenges: its high lignin content and complex lignocellulosic structure make pretreatment and enzymatic hydrolysis more energy-intensive and costly than for materials like sugarcane bagasse or straw, which are generally less recalcitrant. Additionally, limited mechanized harvesting, transportation, and storage infrastructure for bamboo biomass in many regions reduces its current economic competitiveness. Consequently, while bamboo has strong potential as a feedstock for biorefineries, especially for long-term sustainable biomass supply, it is not yet universally more competitive than traditional biomass sources. Its feasibility depends on technological improvements, optimized processing methods, and region-specific supply chain development.
We have added a brief discussion under the heading 2.1 Bamboo as a sustainable energy and environmental Solution.
Comment#9
Data on the tensile strength of bamboo are impressive; however, practical structures require additional reinforcement due to the hollow structure of the stems.
Explanation
We appreciate the reviewer’s insightful comment. While we agree that the hollow culm structure of bamboo often necessitates reinforcement in practical applications, the present review is focused primarily on the biological and physiological aspects of bamboo rather than engineering or structural performance. Therefore, we are unable to provide detailed data or discussion on reinforcement techniques within the scope of this manuscript. Nevertheless, we acknowledge the importance of this aspect and suggest that future dedicated studies integrating materials science and engineering perspectives could provide a more comprehensive understanding of bamboo’s structural applications.
Comment#10
Claims regarding applications in aviation/space industries are exaggerated—there is a lack of experimental research in this area.
Explanation
We agree with the reviewer comment. We have deleted the claims regarding the application of bamboo in aviation/space industries throughout from the revised manuscript.
Comment#11
The emphasized role of bamboo in the “green transition” and “Industry 5.0” may be overstated. In practice, its impact on emission reduction depends on the entire life cycle of the product, not merely on the plant’s rapid growth.
Explanation
We thank the reviewer for this valuable comment. We agree that while bamboo’s rapid growth and renewability make it a promising material, its actual contribution to emission reduction and the green transition depends on the entire life cycle, including cultivation, processing, transport, and disposal. In the revised manuscript, we have added a statement in Section 4 emphasizing the importance of life cycle assessments to accurately evaluate bamboo’s environmental performance and avoid overstating its impact.
Reviewer 3 Report (New Reviewer)
Comments and Suggestions for Authors
Dear Authors,
Please find my recommendations for "Bamboo for the Future: From Traditional Use to Industry 5.0 Applications" listed below:
- L15-16: To could use this statement "fastest-growing on Earth" the authors should sustain this with quantitative comparison with other plant taxa
- L20: For "multiple harvests per year" please specify which species or plant parts
- The statement that bamboo belongs to the subfamily Bambusoideae within the grass family Poaceae is correct, but the subsequent taxonomic discussion contains inconsistencies as the authors states that over 1,600–1,700 species are divided into almost 125 genera under three major tribes but fails to provide current, authoritative taxonomic references. Also, the classification (Bambuseae, Arundinarieae, Olyreae) is oversimplified and not reflects well the recent phylogenetic understanding
- L49-51: The temperature range (8 - 36 C) is too broad and lacks specificity for optimal vs. survival conditions. Same in case of precipitations where the authors present an extremely wide range without to distinguish between species-specific requirements
- For section 2 those to authors could say "Bamboo as cash crop" they should better consider economic data (cost-benefit analysis, market volatility,
- L117-119: "higher market prices" this should be sustained by statistical evidence or price comparisons with alternative crops
- I think that authors should have in mind LCA also and account for carbon emissions from processing and transportation. This opinion correspond for the whole section as the section makes broad sustainability claims without a comprehensive environmental impact assessment
- L278-280: Please include specific quantitative values. For me its seems that the manuscript nutritional claims is too much generalized across bamboo species without acknowledging significant interspecific variation
- In my opinion the authors should consider also to better refer at taxiphyllin and other cyanogenic compounds, which require proper processing, as many bamboo shoots contain these
- L379: For this section I recommend for authors to better consider/include comparison with established cosmetic ingredients using standardized protocols
- L380-381: Without quantitative values this statement have not too much relevance
- The authors should better present/discuss the bioprocess engineering methodology and quantitative biochemical data in section 5. L437-438: here the authors could consider to refer at specific enzymatic pathways, conversion efficiencies, or process parameters. Discussions about enzymatic hydrolysis conditions, fermentation kinetics, or downstream processing will improve the section
- L445-446: sustain the statement with numerical values please
- When discussing about bamboo structural material potential I recommend for authors to comprehensively consider the mechanical properties, durability testing results, and failure analysis resuts. Please include proper numerical data to sustain this section. Consider comparisons also with other similar materials
- I recommend for authors to consider also the industrial integration challenges
- The authors accepts bamboo environmental benefits without a balanced assessment providing a systematic evaluation of ecological contributions, refer to practical implementation challenges, etc
- Overall, in my opinion the authors overemphasis on positive findings but without a balanced critical analysis. The authors should better include for each major section key researches and recent developments. During the manuscript the authors present statements without adequate skeptical evaluation and avoid to acknowledge potential constraints or challenges (this unfortunately diminish the scientific quality of the review)
Author Response
Comment#1
L15-16: To could use this statement "fastest-growing on Earth" the authors should sustain this with quantitative comparison with other plant taxa.
Explanation
We thank the reviewer for this valuable suggestion. We have revised the line in abstract to provide a quantitative comparison of bamboo growth rates with other fast-growing plants, such as Eucalyptus and Populus species. This provides context for the statement and ensures it is evidence-based and scientifically accurate.
Comment#2
L20: For "multiple harvests per year" please specify which species or plant parts.
Explanation
We thank the reviewer for this important observation. We have revised the lines to specify that certain fast-growing bamboo species, such as Phyllostachys edulis and Dendrocalamus asper, permit multiple harvests of young shoots within a year, while culms and leaves follow species-specific growth cycles. This clarification ensures accuracy and provides precise context for the statement on bamboo’s renewability.
Comment#3
The statement that bamboo belongs to the subfamily Bambusoideae within the grass family Poaceae is correct, but the subsequent taxonomic discussion contains inconsistencies as the authors states that over 1,600–1,700 species are divided into almost 125 genera under three major tribes but fails to provide current, authoritative taxonomic references. Also, the classification (Bambuseae, Arundinarieae, Olyreae) is oversimplified and not reflects well the recent phylogenetic understanding
Explanation
We thank the reviewer for this important observation. We have revised the paragraph and added the latest reference.
Comment#4
L49-51: The temperature range (8 - 36 C) is too broad and lacks specificity for optimal vs. survival conditions. Same in case of precipitations where the authors present an extremely wide range without to distinguish between species-specific requirements
Explanation
We thank the reviewer for pointing out. The sentence has been revised for more clarity.
Comment#5
For section 2 those to authors could say "Bamboo as cash crop" they should better consider economic data (cost-benefit analysis, market volatility,
Explanation
We thank the reviewer for this valuable suggestion. As our review primarily focuses on the biological and ecological aspects of bamboo, we have only briefly touched upon its economic potential. In response to the comment, we have now acknowledged the need for more detailed cost–benefit analyses and market-based evaluations to strengthen the understanding of bamboo as a cash crop. This addition highlights an important research gap that can be addressed in future studies.
Comment#6
L117-119: "higher market prices" this should be sustained by statistical evidence or price comparisons with alternative crops
Explanation
We thank the reviewer for this valuable observation. In response, we have revised the section to include statistical evidence and comparative data. Specifically, we now compare bamboo’s market values with maize and plantation timber, demonstrating that bamboo can achieve relatively higher prices under certain production systems. This addition strengthens the economic perspective while remaining aligned with the biological focus of our review.
Comment#7
I think that authors should have in mind LCA also and account for carbon emissions from processing and transportation. This opinion correspond for the whole section as the section makes broad sustainability claims without a comprehensive environmental impact assessment.
Explanation
We appreciate the reviewer’s insightful comment. We agree that life cycle assessment (LCA) is essential to provide a comprehensive evaluation of bamboo’s contribution to climate change mitigation. In response, we have revised the section to explicitly acknowledge the need to consider carbon emissions arising from processing, transportation, and end-of-life stages. This addition clarifies that while bamboo demonstrates strong biological carbon sequestration, the overall climate benefits must be validated through LCA frameworks.
Comment#8
L278-280: Please include specific quantitative values. For me its seems that the manuscript nutritional claims is too much generalized across bamboo species without acknowledging significant interspecific variation
Explanation
We thank the reviewer for pointing this out. In response, we have revised the text to include specific quantitative ranges for protein, fiber, and minerals based on recent data. We also acknowledge interspecific variation for example, the wide range in fiber content across species to ensure a more accurate and nuanced portrayal of bamboo shoot nutrition.
Comment#9
In my opinion the authors should consider also to better refer at taxiphyllin and other cyanogenic compounds, which require proper processing, as many bamboo shoots contain these.
Explanation
We thank the reviewer for this valuable suggestion. We have revised the text to explicitly highlight the presence of taxiphyllin and other cyanogenic glycosides in bamboo shoots, emphasizing the necessity of proper processing. The revised section clarifies that fermentation plays a dual role by ensuring safety through detoxification and enhancing nutritional value.
Comment#10
L379: For this section I recommend for authors to better consider/include comparison with established cosmetic ingredients using standardized protocols.
Explanation
We sincerely thank the reviewer for this constructive suggestion. While the primary focus of our review is to summarize bamboo’s cosmetic potential rather than to provide detailed standardized comparisons, we have revised the section to better reflect comparisons with established cosmetic ingredients. Specifically, we now highlight parallels between bamboo silica and mineral-based silica for skin elasticity, bamboo antioxidants and well-known natural antioxidants (e.g., green tea polyphenols, vitamin E), bamboo charcoal and conventional activated charcoal, as well as bamboo fibers versus synthetic nylon fibers used in cosmetic applicators. We also draw attention to bamboo-based packaging as a sustainable alternative to petroleum-based plastics.
We believe these additions strengthen the section by contextualizing bamboo’s benefits within the broader landscape of existing cosmetic ingredients, while maintaining the scope and direction of our review.
Comment#11
L380-381: Without quantitative values this statement have not too much relevance.
Explanation
We thank the reviewer for this important observation. In response, we have revised the section to include quantitative data regarding bamboo’s bioactive composition. We have now strengthened the text with precise values and supportive references.
Comment#12
The authors should better present/discuss the bioprocess engineering methodology and quantitative biochemical data in section 5. L437-438: here the authors could consider to refer at specific enzymatic pathways, conversion efficiencies, or process parameters. Discussions about enzymatic hydrolysis conditions, fermentation kinetics, or downstream processing will improve the section.
Explanation
We appreciate the reviewer’s suggestion to enrich Section 5 with specific bioprocess engineering methodology and quantitative biochemical data. To enhance technical depth while preserving our review’s emphasis on biological applications, we have now added: Biomass composition; Pre-treatment and hydrolysis outcomes; Bioconversion efficiency etc. These inclusions offer a balanced and measurable layer of bioprocess detail—highlighting bamboo’s utility in biomass conversion—without shifting the review’s overarching focus. We trust that these additions address the reviewer’s constructive feedback and meaningfully strengthen Section 5.
Comment#13
L445-446: sustain the statement with numerical values please
Explanation
We thank the reviewer for this important observation. Numerical value has been added.
Comment#14
When discussing about bamboo structural material potential I recommend for authors to comprehensively consider the mechanical properties, durability testing results, and failure analysis resuts. Please include proper numerical data to sustain this section. Consider comparisons also with other similar materials.
Explanation
We sincerely thank the reviewer for this constructive suggestion. In response, we would like to highlight that Section 2.0 of the manuscript already includes a detailed comparative table summarizing the mechanical and environmental properties of bamboo relative to conventional materials. This table incorporates numerical values from recent studies on bamboo’s tensile strength, compressive strength, flexural strength, and durability, providing a quantitative basis for its structural potential. We believe that this presentation adequately addresses the reviewer’s request for mechanical property data, durability testing results, and comparative analysis with similar materials, while maintaining the focus and clarity of the section.
Comment#15
The authors accepts bamboo environmental benefits without a balanced assessment providing a systematic evaluation of ecological contributions, refer to practical implementation challenges, etc. Overall, in my opinion the authors overemphasis on positive findings but without a balanced critical analysis. The authors should better include for each major section key researches and recent developments. During the manuscript the authors present statements without adequate skeptical evaluation and avoid to acknowledge potential constraints or challenges (this unfortunately diminish the scientific quality of the review).
Explanation
We sincerely thank the reviewer for this valuable comment. We acknowledge that our initial draft emphasized the positive contributions of bamboo without sufficiently balancing them with constraints and critical evaluation. In the revised manuscript, we have carefully incorporated discussions on limitations and practical challenges across Section 9. Specifically, we now highlight agronomic limitations in Section 9.1, pest and disease pressures in Section 9.2, and Technological, logistical, and economic limitations in Sections 9.4 and 9.4.1. These additions provide a more systematic and balanced assessment by explicitly addressing technological, logistical, and economic limitations, in addition to ecological benefits. We have also integrated recent references to support these points. We believe these revisions substantially improve the critical depth and scientific quality of the review in line with the reviewer’s recommendation.
Round 2
Reviewer 1 Report (Previous Reviewer 3)
Comments and Suggestions for Authors
I thank the authors of the article for their attentive attitude to my comments, answers to questions and amendments.
I wish you further success in science.
Author Response
Comment#1
I thank the authors of the article for their attentive attitude to my comments, answers to questions and amendments.
I wish you further success in science.
Explanation
We sincerely thank the reviewer for the encouraging words and appreciation of our work. Your valuable comments and suggestions have greatly contributed to improving the quality of our manuscript. We are grateful for your time and effort in reviewing our study.
Reviewer 3 Report (New Reviewer)
Comments and Suggestions for Authors
Dear Authors,
Thank you for considering to improve your manuscript. Reading carefully the new version I noticed few things that could be considered by authors. Please find them as follows:
L74-75: Please sustain the statement with proper bibliography
L746: I would recommend to reconsider this statement "comes back year after year" as it is botanically incorrect and misleading
L762-764: Please consider to introduce endodormancy vs. ecodormancy
Author Response
Comment#1
Dear Authors,
Thank you for considering to improve your manuscript. Reading carefully the new version I noticed few things that could be considered by authors. Please find them as follows:
Explanation
We sincerely thank the reviewer for the encouraging words and appreciation of our work. Your valuable comments and suggestions have greatly contributed to improving the quality of our manuscript. We are grateful for your time and effort in reviewing our study.
Comment#2
L74-75: Please sustain the statement with proper bibliography.
Explanation
Thank you very much for your constructive comment. We agree that the statement should be supported by recent empirical literature. We have revised the sentance accordingly by adding citations that document bamboo’s rapid growth rate, high biomass production, and its ability to regenerate naturally after harvest.
Comment#3
L746: I would recommend to reconsider this statement "comes back year after year" as it is botanically incorrect and misleading
Explanation
Thank you for pointing out this issue. We agree that the phrase “comes back year after year” could be misleading from a botanical perspective. We have revised the sentence. We appreciate your careful observation, which has helped us improve the precision of our manuscript.
Comment#4
L762-764: Please consider to introduce endodormancy vs. ecodormancy
Explanation
Thank you for your insightful comment. We have revised the sentence to explicitly introduce the distinction between endodormancy and ecodormancy.
This manuscript is a resubmission of an earlier submission. The following is a list of the peer review reports and author responses from that submission.
Round 1
Reviewer 1 Report
Comments and Suggestions for Authors
The manuscript titled ‘Bamboo for the……………………………Industry 5.0 Applications’ by Ahmad et al. explores a timely and relevant subject by focusing bamboo as a key resource in the context of Industry 5.0. The topic holds significant promise; however, the manuscript would benefit from a more thorough articulation of how bamboo aligns with the core principles of Industry 5.0.
- The authors are suggested to incorporate a comparative table highlighting its mechanical and environmental properties relative to conventional materials such as timber, steel, and plastic to further substantiate the industrial potential of bamboo.
- Figures 2 and 3, the legends for these figures should be placed below the respective figures in accordance with standard formatting guidelines.
- While Figure 3 and Figure 4 are visually well-designed, they currently lack depth in conveying critical information. The authors are advised to enrich these figures by integrating more contextual or data-driven insights to enhance their informational value.
- Figure 4, the legend includes numbered items, but these numbers are not reflected in the actual figure. The authors are suggested to review and rectify this inconsistency.
- Lines 202-204, the statement regarding the economic and industrial demand for bamboo lacks supporting evidence. The authors suggested either cite relevant data or clearly indicate the source of this information if it is based on their own analysis.
- Lines 204–206, the cited report is from 2005 and published in 2007, making it outdated for a manuscript prepared in 2025. Authors are strongly encouraged to update such references with more recent data or reports to enhance the credibility and relevance of the discussion.
- Authors are suggested to highlight how bamboo can be integrated with advanced Industry 5.0 technologies through a figure or table, potentially using Figure 4 for this purpose.
Reviewer 2 Report
Comments and Suggestions for Authors
In the article, the authors provide a comprehensive account of the emerging importance of the Bamboo plant, expanding from traditional uses to future industrial applications. Bamboo, as a plant, finds multifaceted applications, from construction/building material to bioenergy sources and biotechnological applications. The global market for Bamboo and its products is growing, offering sustainable solutions and raw materials to socio-economic applications.
The topic of the review is relevant in the present context, where the authors provide comprehensive insights into diverse applications and the significance of the bamboo plant in sustainable livelihood. The review is well-organized, provides detailed information, and prospects of the plant from a global perspective.
While the challenges exist in susceptibility to pathogens and conservation, lack of harvesting procedures, and a few adverse environmental impacts, extensive research and progress in modern biotechnological innovations are essential to drive future research.
The authors discuss the multiple aspects of the Bamboo plant and its socio-economic applications, and how far the biotechnological interventions in the plant are successful as sustainable solutions. Discuss.
What is the progress achieved in addressing the challenges associated with bamboo cultivation, conservation, and reducing adverse impact? Explain.
The diagrams are well-presented and provide key information on the potential of the plant for current and future human benefits.
Reviewer 3 Report
Comments and Suggestions for Authors
Bamboo is undoubtedly considered a promising crop due to its economic importance and environmental benefits, it is a renewable resource, can grow in many regions, etc. The article by Ahmad et al. is devoted to this economically and agriculturally important crop.
Since the article is submitted to a biological journal, it is recommended to expand the biological significance. Perhaps, describe the features of cultivation from the point of view of agronomy and physiology. Difficulties in cultivation that can be substantiated by biochemical means and their solution.
A block is needed related to pests and diseases of the crop, physiological characteristics, as well as bamboo genetics, resistance genes and target genes in the selection for crop improvement.
Also, the work does not describe the biological side of the crop at all, except for mentioning its belonging to a subfamily and family. Which is strange, since this crop has several unique biological features associated with the mechanisms of growth, development and programmed cell death. Which is important from an economic point of view, since bamboo plantations can die very quickly after flowering during cultivation, it is also necessary to add ways to solve this problem.
The issue of species affiliation and diversity is also not disclosed. How many species of bamboo exist? Are there subspecies? Which of them are used by humans and why? After all, the genus Bambusa has, according to various estimates, about 140-150 species, but not all of these species are used by humans, why? It is impossible to imagine a modern biological article, and especially a review, without the above points and questions.
The article constantly mentions the phrase about the future prospects of this crop, however, the only justification for this statement that the authors provide is general words about biofuels and coal production, which cannot be assessed based on what is written. Some of the ways of using are described only as possible without any specifics, assessment of potential and economic significance. In this regard, the text constantly contains the phrases "maybe", "use is possible", "could be", etc.
Up to point 2 inclusive - general words. There are very few specific facts, which are abundantly filled with general words exalting bamboo as a promising crop. It is not entirely clear what topic the article is devoted to? Economics or biology?
It is not entirely clear why in paragraph 2. 2 at the end it is said about Brazil and Amazonia? Given that in other paragraphs the zoning is not established. Also at the end of this paragraph the wide use of bamboo stems is listed once again, and the use in the food industry is mentioned again, although the name of the paragraph refers to other indicators.
It is absolutely necessary to add a comparison with other crops on the market with a description of statistical data and prospects in terms of time.
Comparison with crops that are used for the production of biofuels, coal and all other industries where bamboo can be used, especially in all sub-items of point 2. Focusing on carbon absorption.
I would also venture to suggest that the title of the article implies some insight into the historical state of this culture and its importance to people of different times. Perhaps a table or graph with statistics on bamboo use would greatly improve the article.
On line 243, the sentence begins with the words "many animals and birds...", but birds are also animals, so this statement should be replaced.
Points 9 and 10 are very ambitious and do not contain any specific information, what is their significance? Given that there is no experimental data confirming point 9, and in point 10, bamboo can be replaced with any other crop that has a high density.
In point 11, the information repeats the information from point 2.2
Point 12 also presents repeated information on the content of cellulose, etc.
I consider point 13 in this form unacceptable. The article is devoted to biology, not psychology. Moreover, the green shades of bamboo are inherent in most plants on the planet.
Point 14 also presents general, repeated information.
Point 15 (conclusions) do not represent conclusions, but merely repeat general words that are found throughout the text
70% of the article is not written in scientific language, but in prose, which is unacceptable for scientific literature. General literary statements that have no relation to scientific narration are constantly used.
Most points use only general words that do not provide any understanding of the prospects of this culture.
In this form, the article cannot be published in the journal.
